# Generalized Dantzig Selector: Application to the $k$-support norm

**Soumyadeep Chatterjee**[*]    **Sheng Chen**[*]    **Arindam Banerjee**
Dept. of Computer Science & Engg.
University of Minnesota, Twin Cities
{chatter,shengc,banerjee}@cs.umn.edu

## Abstract

We propose a Generalized Dantzig Selector (GDS) for linear models, in which any norm encoding the parameter structure can be leveraged for estimation. We investigate both computational and statistical aspects of the GDS. Based on conjugate proximal operator, a flexible inexact ADMM framework is designed for solving GDS. Thereafter, non-asymptotic high-probability bounds are established on the estimation error, which rely on Gaussian widths of the unit norm ball and the error set. Further, we consider a non-trivial example of the GDS using $k$-support norm. We derive an efficient method to compute the proximal operator for $k$-support norm since existing methods are inapplicable in this setting. For statistical analysis, we provide upper bounds for the Gaussian widths needed in the GDS analysis, yielding the first statistical recovery guarantee for estimation with the $k$-support norm. The experimental results confirm our theoretical analysis.

## 1  Introduction

The Dantzig Selector (DS) [3, 5] provides an alternative to regularized regression approaches such as Lasso [19, 22] for sparse estimation. While DS does not consider a regularized maximum likelihood approach, [3] has established clear similarities between the estimates from DS and Lasso. While norm regularized regression approaches have been generalized to more general norms [14, 2], the literature on DS has primarily focused on the sparse $L_1$ norm case, with a few notable exceptions which have considered extensions to sparse group-structured norms [11].

In this paper, we consider linear models of the form $\mathbf{y} = \mathbf{X}\boldsymbol{\theta}^* + \mathbf{w}$, where $\mathbf{y} \in \mathbb{R}^n$ is a set of observations, $\mathbf{X} \in \mathbb{R}^{n \times p}$ is a design matrix with i.i.d. standard Gaussian entries, and $\mathbf{w} \in \mathbb{R}^n$ is i.i.d. standard Gaussian noise. For *any* given norm $\mathcal{R}(\cdot)$, the parameter $\boldsymbol{\theta}^*$ is assumed to be structured in terms of having a low value of $\mathcal{R}(\boldsymbol{\theta}^*)$. For this setting, we propose the following Generalized Dantzig Selector (GDS) for parameter estimation:

$$\hat{\boldsymbol{\theta}} = \underset{\boldsymbol{\theta} \in \mathbb{R}^p}{\operatorname{argmin}} \ \mathcal{R}(\boldsymbol{\theta}) \quad \text{s.t. } \mathcal{R}^*\big(\mathbf{X}^T(\mathbf{y} - \mathbf{X}\boldsymbol{\theta})\big) \leq \lambda_p \ , \tag{1}$$

where $\mathcal{R}^*(\cdot)$ is the dual norm of $\mathcal{R}(\cdot)$, and $\lambda_p$ is a suitable constant. If $\mathcal{R}(\cdot)$ is the $L_1$ norm, (1) reduces to standard DS [5]. A key novel aspect of GDS is that the constraint is in terms of the dual norm $\mathcal{R}^*(\cdot)$ of the original structure inducing norm $\mathcal{R}(\cdot)$. It is instructive to contrast GDS with the recently proposed atomic norm based estimation framework [6] which, unlike GDS, considers constraints based on the $L_2$ norm of the error $\|\mathbf{y} - \mathbf{X}\boldsymbol{\theta}\|_2$.

In this paper, we consider both computational and statistical aspects of the GDS. For the $L_1$-norm Dantzig selector, [5] proposed a primal-dual interior point method since the optimization is a linear program. DASSO and its generalization proposed in [10, 9] focused on homotopy methods, which

---

[*]Both authors contributed equally.

provide a piecewise linear solution path through a sequential simplex-like algorithm. However, none of the algorithms above can be immediately extended to our general formulation. In recent work, the Alternating Direction Method of Multipliers (ADMM) has been applied to the $L_1$-norm Dantzig selection problem [12, 21], and the linearized version in [21] proved to be efficient. Motivated by such results for DS, we propose a general inexact ADMM [20] framework for GDS where the primal update steps, interestingly, turn out respectively to be proximal updates involving $\mathcal{R}(\boldsymbol{\theta})$ and its convex conjugate, the indicator of $\mathcal{R}^*(\mathbf{x}) \leq \lambda_p$. As a result, by Moreau decomposition, it suffices to develop efficient proximal update for either $\mathcal{R}(\boldsymbol{\theta})$ or its conjugate. On the statistical side, we establish non-asymptotic high-probability bounds on the estimation error $\|\hat{\boldsymbol{\theta}} - \boldsymbol{\theta}^*\|_2$. Interestingly, the bound depends on the Gaussian width of the unit norm ball of $\mathcal{R}(\cdot)$ as well as the Gaussian width of intersection of error cone and unit sphere [6, 16].

As a non-trivial example of the GDS framework, we consider estimation using the recently proposed $k$-support norm [1, 13]. We show that proximal operators for $k$-support norm can be efficiently computed in $O(p \log p + \log k \log(p - k))$, and hence the estimation can be done efficiently. Note that existing work [1, 13] on $k$-support norm has focused on the proximal operator for the *square* of the $k$-support norm, which is not directly applicable in our setting. On the statistical side, we provide upper bounds for the Gaussian widths of the unit norm ball and the error cone as needed in the GDS framework, yielding the first statistical recovery guarantee for estimation with the $k$-support norm.

The rest of the paper is organized as follows: We establish general optimization and statistical recovery results for GDS for any norm in Section 2. In Section 3, we present efficient algorithms and estimation error bounds for the $k$-support norm. We present experimental results in Section 4 and conclude in Section 5. All technical analysis and proofs can be found in [7].

## 2 General Optimization and Statistical Recovery Guarantees

The problem in (1) is a convex program, and a suitable choice of $\lambda_p$ ensures that the feasible set is not empty. We start the section with an inexact ADMM framework for solving problems of the form (1), and then present bounds on the estimation error establishing statistical consistency of GDS.

### 2.1 General Optimization Framework using Inexact ADMM

For convenience, we temporarily drop the subscript $p$ of $\lambda_p$. We let $\mathbf{A} = \mathbf{X}^T \mathbf{X}$, $\mathbf{b} = \mathbf{X}^T \mathbf{y}$, and define the set $\mathcal{C}_\lambda = \{\mathbf{v} : \mathcal{R}^*(\mathbf{v}) \leq \lambda\}$. The optimization problem is equivalent to

$$\min_{\boldsymbol{\theta}, \mathbf{v}} \mathcal{R}(\boldsymbol{\theta}) \quad \text{s.t. } \mathbf{b} - \mathbf{A}\boldsymbol{\theta} = \mathbf{v}, \ \mathbf{v} \in \mathcal{C}_\lambda \,. \tag{2}$$

Due to the nonsmoothness of both $\mathcal{R}$ and $\mathcal{R}^*$, solving (2) can be quite challenging and a generally applicable algorithm is Alternating Direction Method of Multipliers (ADMM) [4]. The augmented Lagrangian function for (2) is given as

$$\mathcal{L}_{\mathcal{R}}(\boldsymbol{\theta}, \mathbf{v}, \mathbf{z}) = \mathcal{R}(\boldsymbol{\theta}) + \langle \mathbf{z}, \mathbf{A}\boldsymbol{\theta} + \mathbf{v} - \mathbf{b} \rangle + \frac{\rho}{2}\|\mathbf{A}\boldsymbol{\theta} + \mathbf{v} - \mathbf{b}\|_2^2 \,, \tag{3}$$

where $\mathbf{z}$ is the Lagrange multiplier and $\rho$ controls the penalty introduced by the quadratic term. The iterative updates of the variables $(\boldsymbol{\theta}, \mathbf{v}, \mathbf{z})$ in standard ADMM are given by

$$\boldsymbol{\theta}^{k+1} \leftarrow \operatorname*{argmin}_{\boldsymbol{\theta}} \mathcal{L}_{\mathcal{R}}(\boldsymbol{\theta}, \mathbf{v}^k, \mathbf{z}^k) \,, \tag{4}$$

$$\mathbf{v}^{k+1} \leftarrow \operatorname*{argmin}_{\mathbf{v} \in \mathcal{C}_\lambda} \mathcal{L}_{\mathcal{R}}(\boldsymbol{\theta}^{k+1}, \mathbf{v}, \mathbf{z}^k) \,, \tag{5}$$

$$\mathbf{z}^{k+1} \leftarrow \mathbf{z}^k + \rho(\mathbf{A}\boldsymbol{\theta}^{k+1} + \mathbf{v}^{k+1} - \mathbf{b}) \,. \tag{6}$$

Note that update (4) amounts to a norm regularized least squares problem for $\boldsymbol{\theta}$, which can be computationally expensive. Thus we use an inexact update for $\boldsymbol{\theta}$ instead, which can alleviate the computational cost and lead to a quite simple algorithm. Inspired by [21, 20], we consider a simpler subproblem for the $\boldsymbol{\theta}$-update which minimizes

$$\begin{aligned} \widetilde{\mathcal{L}}_{\mathcal{R}}^k(\boldsymbol{\theta}, \mathbf{v}^k, \mathbf{z}^k) = {} & \mathcal{R}(\boldsymbol{\theta}) + \langle \mathbf{z}^k, \mathbf{A}\boldsymbol{\theta} + \mathbf{v}^k - \mathbf{b} \rangle + \frac{\rho}{2}\Big(\|\mathbf{A}\boldsymbol{\theta}^k + \mathbf{v}^k - \mathbf{b}\|_2^2 + \\ & 2\langle \boldsymbol{\theta} - \boldsymbol{\theta}^k, \mathbf{A}^T(\mathbf{A}\boldsymbol{\theta}^k + \mathbf{v}^k - \mathbf{b}) \rangle + \frac{\mu}{2}\|\boldsymbol{\theta} - \boldsymbol{\theta}^k\|_2^2 \Big) \,, \end{aligned} \tag{7}$$

---
**Algorithm 1** ADMM for Generalized Dantzig Selector
---
**Input:** $\mathbf{A} = \mathbf{X}^T\mathbf{X}$, $\mathbf{b} = \mathbf{X}^T\mathbf{y}$, $\rho$, $\mu$
**Output:** Optimal $\hat{\theta}$ of (1)
 1: Initialize $(\boldsymbol{\theta}, \mathbf{v}, \mathbf{z})$
 2: **while** not converged **do**
 3:    $\boldsymbol{\theta}^{k+1} \leftarrow \mathbf{prox}_{\frac{2\mathcal{R}}{\rho\mu}}\left(\boldsymbol{\theta}^k - \frac{2}{\mu}\mathbf{A}^T(\mathbf{A}\boldsymbol{\theta}^k + \mathbf{v}^k - \mathbf{b} + \frac{\mathbf{z}^k}{\rho})\right)$
 4:    $\mathbf{v}^{k+1} \leftarrow \mathbf{prox}_{\mathbb{I}_{\mathcal{C}_\lambda}}\left(\mathbf{b} - \mathbf{A}\boldsymbol{\theta}^{k+1} - \frac{\mathbf{z}^k}{\rho}\right)$
 5:    $\mathbf{z}^{k+1} \leftarrow \mathbf{z}^k + \rho(\mathbf{A}\boldsymbol{\theta}^{k+1} + \mathbf{v}^{k+1} - \mathbf{b})$
 6: **end while**
---

where $\mu$ is a user-defined parameter. $\widetilde{\mathcal{L}}_{\mathcal{R}}^k(\boldsymbol{\theta}, \mathbf{v}^k, \mathbf{z}^k)$ can be viewed as an approximation of $\mathcal{L}_{\mathcal{R}}(\boldsymbol{\theta}, \mathbf{v}^k, \mathbf{z}^k)$ with the quadratic term linearized at $\boldsymbol{\theta}^k$. Then the update (4) is replaced by

$$
\begin{aligned}
\boldsymbol{\theta}^{k+1} &\leftarrow \underset{\boldsymbol{\theta}}{\operatorname{argmin}} \, \widetilde{\mathcal{L}}_{\mathcal{R}}^k(\boldsymbol{\theta}, \mathbf{v}^k, \mathbf{z}^k) \\
&= \underset{\boldsymbol{\theta}}{\operatorname{argmin}} \left\{ \frac{2\mathcal{R}(\boldsymbol{\theta})}{\rho\mu} + \frac{1}{2}\left\| \boldsymbol{\theta} - \left(\boldsymbol{\theta}^k - \frac{2}{\mu}\mathbf{A}^T(\mathbf{A}\boldsymbol{\theta}^k + \mathbf{v}^k - \mathbf{b} + \frac{\mathbf{z}^k}{\rho})\right) \right\|_2^2 \right\}.
\end{aligned} \tag{8}
$$

Similarly the update of $\mathbf{v}$ in (5) can be recast as

$$
\mathbf{v}^{k+1} \leftarrow \underset{\mathbf{v} \in \mathcal{C}_\lambda}{\operatorname{argmin}} \, \mathcal{L}_{\mathcal{R}}(\boldsymbol{\theta}^{k+1}, \mathbf{v}, \mathbf{z}^k) = \underset{\mathbf{v} \in \mathcal{C}_\lambda}{\operatorname{argmin}} \, \frac{1}{2}\left\| \mathbf{v} - (\mathbf{b} - \mathbf{A}\boldsymbol{\theta}^{k+1} - \frac{\mathbf{z}^k}{\rho}) \right\|_2^2. \tag{9}
$$

In fact, the updates of both $\boldsymbol{\theta}$ and $\mathbf{v}$ are to compute certain *proximal operators* [15]. In general, the proximal operator $\mathbf{prox}_h(\cdot)$ of a closed proper convex function $h : \mathbb{R}^p \to \mathbb{R} \cup \{+\infty\}$ is defined as

$$
\mathbf{prox}_h(\mathbf{x}) = \underset{\mathbf{w} \in \mathbb{R}^p}{\operatorname{argmin}} \left\{ \frac{1}{2}\|\mathbf{w} - \mathbf{x}\|_2^2 + h(\mathbf{w}) \right\}.
$$

Hence it is easy to see that (8) and (9) correspond to $\mathbf{prox}_{\frac{2\mathcal{R}}{\rho\mu}}(\cdot)$ and $\mathbf{prox}_{\mathbb{I}_{\mathcal{C}_\lambda}}(\cdot)$, respectively, where $\mathbb{I}_{\mathcal{C}_\lambda}(\cdot)$ is the indicator function of set $\mathcal{C}_\lambda$ given by

$$
\mathbb{I}_{\mathcal{C}_\lambda}(\mathbf{x}) = \begin{cases} 0 & \text{if } \mathbf{x} \in \mathcal{C}_\lambda \\ +\infty & \text{if otherwise} \end{cases}.
$$

In Algorithm 1, we provide our general ADMM for the GDS. For the ADMM to work, we need two subroutines that can efficiently compute the proximal operators for the functions in Line 3 and 4 respectively. The simplicity of the proposed approach stems from the fact that we in fact need *only one* subroutine, for any one of the functions, since the functions are conjugates of each other.

**Proposition 1** *Given $\beta > 0$ and a norm $\mathcal{R}(\cdot)$, the two functions, $f(\mathbf{x}) = \beta\mathcal{R}(\mathbf{x})$ and $g(\mathbf{x}) = \mathbb{I}_{\mathcal{C}_\beta}(\mathbf{x})$ are convex conjugate to each other, thus giving the following identity,*

$$
\mathbf{x} = \mathbf{prox}_f(\mathbf{x}) + \mathbf{prox}_g(\mathbf{x}). \tag{10}
$$

*Proof:* The Proposition 1 simply follows from the definition of convex conjugate and dual norm, and (10) is just *Moreau decomposition* provided in [15]. ∎

The decomposition enables conversion of the two types of proximal operator to each other at negligible cost (i.e., vector subtraction). Thus we have the flexibility in Algorithm 1 to focus on the proximal operator that is efficiently computable, and the other can be simply obtained through (10).

**Remark on convergence:** Note that Algorithm 1 is a special case of inexact Bregman ADMM proposed in [20], which matches the case of linearizing quadratic penalty term by using $B_{\varphi'_{\boldsymbol{\theta}}}(\boldsymbol{\theta}, \boldsymbol{\theta}_k) = \frac{1}{2}\|\boldsymbol{\theta} - \boldsymbol{\theta}_k\|_2^2$ as Bregman divergence. In order to converge, the algorithm requires $\frac{\mu}{2}$ to be larger than the spectral radius of $\mathbf{A}^T\mathbf{A}$, and the convergence rate is $O(1/T)$ according to Theorem 2 in [20].

## 2.2 Statistical Recovery for Generalized Dantzig Selector

Our goal is to provide non-asymptotic bounds on $\|\hat{\boldsymbol{\theta}} - \boldsymbol{\theta}^*\|_2$ between the true parameter $\boldsymbol{\theta}^*$ and the minimizer $\hat{\boldsymbol{\theta}}$ of (1). Let the error vector be defined as $\hat{\Delta} = \hat{\boldsymbol{\theta}} - \boldsymbol{\theta}^*$. For any set $\Omega \subseteq \mathbb{R}^p$, we would measure the size of this set using its Gaussian width [17, 6], which is defined as $\omega(\Omega) = \mathbf{E_g}\left[\sup_{\mathbf{z}\in\Omega}\langle\mathbf{g},\mathbf{z}\rangle\right]$, where $\mathbf{g}$ is a vector of i.i.d. standard Gaussian entries. We also consider the error cone $\mathcal{T}_{\mathcal{R}}(\boldsymbol{\theta}^*)$, generated by the set of possible error vectors $\Delta$ and containing $\hat{\Delta}$, defined as

$$\mathcal{T}_{\mathcal{R}}(\boldsymbol{\theta}^*) := \text{cone}\{\Delta \in \mathbb{R}^p \; : \; \mathcal{R}(\boldsymbol{\theta}^* + \Delta) \leq \mathcal{R}(\boldsymbol{\theta}^*)\} \; . \tag{11}$$

Note that this set contains a restricted set of directions and does not in general span the entire space of $\mathbb{R}^p$. With these definitions, we obtain our main result.

**Theorem 1** *Suppose that both design matrix $\mathbf{X}$ and noise $\mathbf{w}$ consists of i.i.d. Gaussian entries with zero mean variance 1 and $\mathbf{X}$ has normalized columns, i.e. $\|\mathbf{X}^{(j)}\|_2 = 1$, $j = 1, \ldots, p$. If we solve the problem (1) with*

$$\lambda_p \geq c\mathbf{E}\left[\mathcal{R}^*(\mathbf{X}^T\mathbf{w})\right] \; , \tag{12}$$

*where $c > 1$ is a constant, then, with probability at least $(1 - \eta_1 \exp(-\eta_2 n))$, we have*

$$\|\hat{\boldsymbol{\theta}} - \boldsymbol{\theta}^*\|_2 \leq \frac{4\sqrt{\mathcal{R}(\boldsymbol{\theta}^*)\lambda_p}}{(\ell_n - \omega(\mathcal{T}_{\mathcal{R}}(\boldsymbol{\theta}^*) \cap \mathbb{S}^{p-1}))} \; , \tag{13}$$

*where $\omega(\mathcal{T}_{\mathcal{R}}(\boldsymbol{\theta}^*)\cap\mathbb{S}^{p-1})$ is the Gaussian width of the intersection of the error cone $\mathcal{T}_{\mathcal{R}}(\boldsymbol{\theta}^*)$ and the unit spherical shell $\mathbb{S}^{p-1}$, and $\ell_n$ is the expected length of a length $n$ i.i.d. standard Gaussian vector with $\frac{n}{\sqrt{n+1}} < \ell_n < \sqrt{n}$, and $\eta_1, \eta_2 > 0$ are constants.*

**Remark:** The choice of $\lambda_p$ is also intimately connected to the notion of Gaussian width. Note that for $\mathbf{X}$ with unit length columns, $\mathbf{X}^T\mathbf{w} = \mathbf{z}$ is an i.i.d. standard Gaussian vector. Therefore the right hand side of (12) can be written as

$$\mathbf{E}\left[\mathcal{R}^*(\mathbf{X}^T\mathbf{w})\right] = \mathbf{E}\left[\sup_{\mathbf{u}: \, \mathcal{R}(\mathbf{u})\leq 1}\langle\mathbf{u},\mathbf{z}\rangle\right] = \omega\left(\{\mathbf{u} \; : \; \mathcal{R}(\mathbf{u}) \leq 1\}\right) \; , \tag{14}$$

which is the Gaussian width of the unit ball of the norm $\mathcal{R}(\cdot)$.

**Example: $L_1$-norm Dantzig Selector** When $\mathcal{R}(\cdot)$ is chosen to be $L_1$ norm, the dual norm is the $L_\infty$ norm, and (1) is reduced to the standard DS, given by

$$\hat{\boldsymbol{\theta}} = \operatorname*{argmin}_{\boldsymbol{\theta}\in\mathbb{R}^p} \|\boldsymbol{\theta}\|_1 \quad \text{s.t. } \|\mathbf{X}^T(\mathbf{y} - \mathbf{X}\boldsymbol{\theta})\|_\infty \leq \lambda \; . \tag{15}$$

We know that $\mathbf{prox}_{\beta\|\cdot\|_1}(\cdot)$ is given by the elementwise soft-thresholding operation

$$\left[\mathbf{prox}_{\beta\|\cdot\|_1}(\mathbf{x})\right]_i = \text{sign}(\mathbf{x}_i) \cdot \max(0, |\mathbf{x}_i| - \beta) \; . \tag{16}$$

Based on Proposition 1, the ADMM updates in Algorithm 1 can be instantiated as

$$\boldsymbol{\theta}^{k+1} \leftarrow \mathbf{prox}_{\frac{2\|\cdot\|_1}{\rho\mu}}\left(\boldsymbol{\theta}^k - \frac{2}{\mu}\mathbf{A}^T(\mathbf{A}\boldsymbol{\theta}^k + \mathbf{v}^k - \mathbf{u} + \frac{\mathbf{z}^k}{\rho})\right) \; ,$$

$$\mathbf{v}^{k+1} \leftarrow (\mathbf{u} - \mathbf{A}\boldsymbol{\theta}^{k+1} - \frac{\mathbf{z}^k}{\rho}) - \mathbf{prox}_{\lambda\|\cdot\|_1}\left(\mathbf{u} - \mathbf{A}\boldsymbol{\theta}^{k+1} - \frac{\mathbf{z}^k}{\rho}\right) \; ,$$

$$\mathbf{z}^{k+1} \leftarrow \mathbf{z}^k + \rho(\mathbf{A}\boldsymbol{\theta}^{k+1} + \mathbf{v}^{k+1} - \mathbf{u}) \; ,$$

where the update of $\mathbf{v}$ leverages the decomposition (10). Similar updates were used in [21] for $L_1$-norm Dantzig selector.

For statistical recovery, we assume that $\boldsymbol{\theta}^*$ is $s$-sparse, i.e., contains $s$ non-zero entries, and that $\|\boldsymbol{\theta}^*\|_2 = 1$, so that $\|\boldsymbol{\theta}^*\|_1 \leq s$. It was shown in [6] that the Gaussian width of the set $(\mathcal{T}_{L_1}(\boldsymbol{\theta}^*) \cap \mathbb{S}^{p-1})$ is upper bounded as $\omega(\mathcal{T}_{L_1}(\boldsymbol{\theta}^*)\cap\mathbb{S}^{p-1})^2 \leq 2s\log\left(\frac{p}{s}\right) + \frac{5}{4}s$. Also note that $\mathbf{E}\left[\mathcal{R}^*(\mathbf{X}^T\mathbf{w})\right] =$

$\mathbf{E}[\|\mathbf{X}^T\mathbf{w}\|_\infty] \leq \log p$, since $\mathbf{X}^T\mathbf{w}$ is a vector of i.i.d. standard Gaussian entries [5]. Therefore, if we solve (15) with $\lambda_p = 2\log p$, then

$$\|\hat{\boldsymbol{\theta}} - \boldsymbol{\theta}^*\|_2 \leq \frac{\sqrt{32\|\boldsymbol{\theta}^*\|_1 \log p}}{\left(\ell_n - \sqrt{2s\log\left(\frac{p}{s}\right) + \frac{5}{4}s}\right)} = \mathbf{O}\left(\sqrt{\frac{s\log p}{n}}\right) \tag{17}$$

with high probability, which agrees with known results for DS [3, 5].

## 3 Dantzig Selection with $k$-support norm

We first introduce some notations. Given any $\boldsymbol{\theta} \in \mathbb{R}^p$, let $|\boldsymbol{\theta}|$ denote its absolute-valued counterpart and $\boldsymbol{\theta}^\downarrow$ denote the permutation of $\boldsymbol{\theta}$ with its elements arranged in decreasing order. In previous work [1, 13], the $k$-support norm has been defined as

$$\|\boldsymbol{\theta}\|_k^{sp} = \min\left\{\sum_{I \in \mathcal{G}^{(k)}} \|v_I\|_2 \ : \ \text{supp}(v_I) \subseteq I, \ \sum_{I \in \mathcal{G}^{(k)}} v_I = \boldsymbol{\theta}\right\}, \tag{18}$$

where $\mathcal{G}^{(k)}$ denotes the set of subsets of $\{1,\ldots,p\}$ of cardinality at most $k$. The unit ball of this norm is the set $C_k = \text{conv}\left\{\boldsymbol{\theta} \in \mathbb{R}^p \ : \ \|\boldsymbol{\theta}\|_0 \leq k, \|\boldsymbol{\theta}\|_2 \leq 1\right\}$. The dual norm of the $k$-support norm is given by

$$\|\boldsymbol{\theta}\|_k^{sp^*} = \max\left\{\|\boldsymbol{\theta}_G\|_2 \ : \ G \in \mathcal{G}^{(k)}\right\} = \left(\sum_{i=1}^k |\boldsymbol{\theta}|_i^{\downarrow^2}\right)^{\frac{1}{2}}. \tag{19}$$

Note that $k = 1$ gives the $L_1$ norm and its dual norm is $L_\infty$ norm. The $k$-support norm was proposed in order to overcome some of the empirical shortcomings of the elastic net [23] and the (group)-sparse regularizers. It was shown in [1] to behave similarly as the elastic net in the sense that the unit norm ball of the $k$-support norm is within a constant factor of $\sqrt{2}$ of the unit elastic net ball. Although multiple papers have reported good empirical performance of the $k$-support norm on selecting correlated features, where $L_1$ regularization fails, there exists no statistical analysis of the $k$-support norm. Besides, current computational methods consider square of $k$-support norm in their formulation, which might fail to work out in certain cases.

In the rest of this section, we focus on GDS with $\mathcal{R}(\boldsymbol{\theta}) = \|\boldsymbol{\theta}\|_k^{sp}$ given as

$$\hat{\boldsymbol{\theta}} = \underset{\boldsymbol{\theta} \in \mathbb{R}^p}{\text{argmin}} \|\boldsymbol{\theta}\|_k^{sp} \qquad \text{s.t.} \quad \|\mathbf{X}^T(\mathbf{y} - \mathbf{X}\boldsymbol{\theta})\|_k^{sp^*} \leq \lambda_p . \tag{20}$$

For the indicator function $\mathbb{I}_{\mathcal{C}_\lambda}(\cdot)$ of the dual norm, we present a fast algorithm for computing its proximal operator by exploiting the structure of its solution, which can be directly plugged in Algorithm 1 to solve (20). Further, we prove statistical recovery bounds for $k$-support norm Dantzig selection, which hold even for a high-dimensional scenario, where $n < p$.

### 3.1 Computation of Proximal Operator

In order to solve (20), either $\mathbf{prox}_{\lambda\|\cdot\|_k^{sp}}(\cdot)$ or $\mathbf{prox}_{\mathbb{I}_{\mathcal{C}_\lambda}}(\cdot)$ for $\|\cdot\|_k^{sp^*}$ should be efficiently computable. Existing methods [1, 13] are inapplicable to our scenario since they compute the proximal operator for squared $k$-support norm, from which $\mathbf{prox}_{\mathbb{I}_{\mathcal{C}_\lambda}}(\cdot)$ cannot be directly obtained. In Theorem 2, we show that $\mathbf{prox}_{\mathbb{I}_{\mathcal{C}_\lambda}}(\cdot)$ can be efficiently computed, and thus Algorithm 1 is applicable.

**Theorem 2** *Given $\lambda > 0$ and $\mathbf{x} \in \mathbb{R}^p$, if $\|\mathbf{x}\|_k^{sp^*} \leq \lambda$, then $\mathbf{w}^* = \mathbf{prox}_{\mathbb{I}_{\mathcal{C}_\lambda}}(\mathbf{x}) = \mathbf{x}$. If $\|\mathbf{x}\|_k^{sp^*} > \lambda$, define $A_{sr} = \sum_{i=s+1}^r |\mathbf{x}|_i^\downarrow$, $B_s = \sum_{i=1}^s (|\mathbf{x}|_i^\downarrow)^2$, in which $0 \leq s < k$ and $k \leq r \leq p$, and construct the nonlinear equation of $\beta$,*

$$(k-s)A_{sr}^2\left[\frac{1+\beta}{r-s+(k-s)\beta}\right]^2 - \lambda^2(1+\beta)^2 + B_s = 0 . \tag{21}$$

*Let $\beta_{sr}$ be given by*

$$\beta_{sr} = \begin{cases} \textit{nonnegative root of (21)} & \textit{if } s > 0 \textit{ and the root exists} \\ 0 & \textit{otherwise} \end{cases} . \tag{22}$$

*Then the proximal operator $\mathbf{w}^* = \mathbf{prox}_{\mathbb{I}_{\mathcal{C}_\lambda}}(\mathbf{x})$ is given by*

$$|\mathbf{w}^*|_i^{\downarrow} = \begin{cases} \frac{1}{1+\beta_{s^*r^*}}|\mathbf{x}|_i^{\downarrow} & \textit{if } 1 \leq i \leq s^* \\ \sqrt{\frac{\lambda^2 - B_{s^*}}{k-s^*}} & \textit{if } s^* < i \leq r^* \textit{ and } \beta_{s^*r^*} = 0 \\ \frac{A_{s^*r^*}}{r^* - s^* + (k-s^*)\beta_{s^*r^*}} & \textit{if } s^* < i \leq r^* \textit{ and } \beta_{s^*r^*} > 0 \\ |\mathbf{x}|_i^{\downarrow} & \textit{if } r^* < i \leq p \end{cases} , \tag{23}$$

*where the indices $s^*$ and $r^*$ with computed $|\mathbf{w}^*|^{\downarrow}$ satisfy the following two inequalities:*

$$|\mathbf{w}^*|_{s^*}^{\downarrow} > |\mathbf{w}^*|_k^{\downarrow} , \tag{24}$$

$$|\mathbf{x}|_{r^*+1}^{\downarrow} \leq |\mathbf{w}^*|_k^{\downarrow} < |\mathbf{x}|_{r^*}^{\downarrow} . \tag{25}$$

*There might be multiple pairs of $(s,r)$ satisfying the inequalities (24)-(25), and we choose the pair with the smallest $\||\mathbf{x}|^{\downarrow} - |\mathbf{w}^*|^{\downarrow}\|_2$. Finally, $\mathbf{w}^*$ is obtained by sign-changing and reordering $|\mathbf{w}^*|^{\downarrow}$ to conform to $\mathbf{x}$.*

**Remark:** The nonlinear equation (21) is quartic, for which we can use general formula to get all the roots [18]. In addition, if it exists, the nonnegative root is unique, as shown in the proof [7].

Theorem 2 indicates that computing $\mathbf{prox}_{\mathbb{I}_{\mathcal{C}_\lambda}}(\cdot)$ requires sorting of entries in $|\mathbf{x}|$ and a two-dimensional search of $s^*$ and $r^*$. Hence the total time complexity is $O(p \log p + k(p-k))$. However, a more careful observation can particularly reduce the search complexity from $O(k(p-k))$ to $O(\log k \log(p-k))$, which is motivated by Theorem 3.

**Theorem 3** *In search of $(s^*, r^*)$ defined in Theorem 2, there can be only one $\tilde{r}$ for a given candidate $\tilde{s}$ of $s^*$, such that the inequality (25) is satisfied. Moreover if such $\tilde{r}$ exists, then for any $r < \tilde{r}$, the associated $|\tilde{\mathbf{w}}|_k^{\downarrow}$ violates the first part of (25), and for $r > \tilde{r}$, $|\tilde{\mathbf{w}}|_k^{\downarrow}$ violates the second part of (25). On the other hand, based on the $\tilde{r}$, we have following assertion of $s^*$,*

$$s^* \begin{cases} > \tilde{s} & \textit{if } \tilde{r} \textit{ does not exist} \\ \geq \tilde{s} & \textit{if } \tilde{r} \textit{ exists and corresponding } |\tilde{\mathbf{w}}|_k^{\downarrow} \textit{ satisfies (24)} \\ < \tilde{s} & \textit{if } \tilde{r} \textit{ exists but corresponding } |\tilde{\mathbf{w}}|_k^{\downarrow} \textit{ violates (24)} \end{cases} . \tag{26}$$

Based on Theorem 3, the accelerated search procedure for finding $(s^*, r^*)$ is to execute a two-dimensional binary search, and Algorithm 2 gives the details. Therefore the total time complexity becomes $O(p \log p + \log k \log(p-k))$. Compared with previous proximal operators for squared $k$-support norm, this complexity is better than that in [1], and roughly the same as the one in [13].

### 3.2 Statistical Recovery Guarantees for $k$-support norm

The analysis of the generalized Dantzig Selector for $k$-support norm consists of addressing two key challenges. First, note that Theorem 1 requires an appropriate choice of $\lambda_p$. Second, one needs to compute the Gaussian width of the subset of the error set $\mathcal{T}_{\mathcal{R}}(\boldsymbol{\theta}^*) \cap \mathbb{S}^{p-1}$. For the $k$-support norm, we can get upper bounds to both of these quantities. We start by defining some notation. Let $\mathcal{G}^* \subseteq \mathcal{G}^{(k)}$ be the set of groups intersecting with the support of $\boldsymbol{\theta}^*$, and let $S$ be the union of groups in $\mathcal{G}^*$, such that $s = |S|$. Then, we have the following bounds which are used for choosing $\lambda_p$, and bounding the Gaussian width.

**Theorem 4** *For the $k$-support norm Generalized Dantzig Selection problem (20), we obtain*

$$\mathbf{E}\left[\mathcal{R}^*(\mathbf{X}^T\mathbf{w})\right] \leq k \left(\sqrt{2 \log \left(\frac{ep}{k}\right)} + 1\right)^2 \tag{27}$$

$$\omega(\mathcal{T}_{\mathcal{R}}(\boldsymbol{\theta}^*) \cap \mathbb{S}^{p-1})^2 \leq \left(\sqrt{2k \log \left(p - k - \left\lceil\frac{s}{k}\right\rceil + 2\right)} + \sqrt{k}\right)^2 \cdot \left\lceil\frac{s}{k}\right\rceil + s . \tag{28}$$

**Algorithm 2** Algorithm for computing $\mathbf{prox}_{\mathbb{I}_{C_\lambda}}(\cdot)$ of $\|\cdot\|_k^{sp^*}$

---

**Input:** $\mathbf{x}, k, \lambda$
**Output:** $\mathbf{w}^* = \mathbf{prox}_{\mathbb{I}_{C_\lambda}}(\mathbf{x})$
1: **if** $\|\mathbf{x}\|_k^{sp^*} \leq \lambda$ **then**
2: $\quad \mathbf{w}^* := \mathbf{x}$
3: **else**
4: $\quad l := 0$, $u := k - 1$, and sort $|\mathbf{x}|$ to get $|\mathbf{x}|^{\downarrow}$
5: $\quad$ **while** $l \leq u$ **do**
6: $\quad\quad \tilde{s} := \lfloor (l+u)/2 \rfloor$, and binary search for $\tilde{r}$ that satisfies (25) and compute $\tilde{\mathbf{w}}$ based on (23)
7: $\quad\quad$ **if** $\tilde{r}$ does not exist **then**
8: $\quad\quad\quad l := \tilde{s} + 1$
9: $\quad\quad$ **else if** $\tilde{r}$ exists and (24) is satisfied **then**
10: $\quad\quad\quad \mathbf{w}^* := \tilde{\mathbf{w}}$, $l := \tilde{s} + 1$
11: $\quad\quad$ **else if** $\tilde{r}$ exists but (24) is not satisfied **then**
12: $\quad\quad\quad u := \tilde{s} - 1$
13: $\quad\quad$ **end if**
14: $\quad$ **end while**
15: **end if**

---

Our analysis technique for these bounds follows [16]. Similar results were obtained in [8] in the context of calculating norms of Gaussian vectors, and our bounds are of the same order as those of [8]. Choosing $\lambda_p = 2k\left(\sqrt{2\log\left(\frac{ep}{k}\right)} + 1\right)^2$, and under the assumptions of Theorem 1, we obtain the following result on the error bound for the minimizer of (20).

**Corollary 1** *Suppose that we obtain i.i.d. Gaussian measurements* $\mathbf{X}$, *and the noise* $\mathbf{w}$ *is i.i.d.* $\mathcal{N}(0,1)$. *If we solve* (20) *with* $\lambda_p$ *chosen as above. Then, with high probability, we obtain*

$$\|\hat{\boldsymbol{\theta}} - \boldsymbol{\theta}^*\|_2 \leq \frac{\sqrt{8\|\boldsymbol{\theta}^*\|_k^{sp}}\left(\sqrt{2k\log\left(\frac{ep}{k}\right)} + \sqrt{k}\right)}{(\ell_n - \omega(\mathcal{T}_\mathcal{R}(\boldsymbol{\theta}^*) \cap \mathbb{S}^{p-1}))} = O\left(\frac{\sqrt{sk\log\left(\frac{p}{k}\right)} + \sqrt{sk}}{\sqrt{n}}\right). \quad (29)$$

**Remark** The error bound provides a natural interpretation for the two special cases of the $k$-support norm, viz. $k = 1$ and $k = p$. First, for $k = 1$ the $k$-support norm is exactly the same as the $L_1$ norm, and the error bound obtained will be $O\left(\sqrt{\frac{s\log p}{n}}\right)$, the same as known results of DS, and shown in Section 2.2. Second, for $k = p$, the $k$-support norm is equal to the $L_2$ norm, and the error cone (11) is then simply a half space (there is no structural constraint) and the error bound scales as $O\left(\sqrt{\frac{sp}{n}}\right)$.

## 4 Experimental Results

On the optimization side, we focus on the efficiency of different proximal operators related to $k$-support norm. On the statistical side, we concentrate on the behavior and performance of GDS with $k$-support norm. All experiments are implemented in MATLAB.

### 4.1 Efficiency of Proximal Operator

We tested four proximal operators related to $k$-support norm, which are normal $\mathbf{prox}_{\mathbb{I}_{C_\lambda}}(\cdot)$ in Theorem 2 and the accelerated one in Theorem 3, $\mathbf{prox}_{\frac{1}{2\beta}(\|\cdot\|_k^{sp})^2}(\cdot)$ in [1], and $\mathbf{prox}_{\frac{\lambda}{2}\|\cdot\|_\Theta^2}(\cdot)$ in [13]. The dimension $p$ of vector varied from 1000 to 10000, and the ratio $p/k = \{200, 100, 50, 20\}$. As illustrated in Figure 1, the speedup of accelerated $\mathbf{prox}_{\mathbb{I}_{C_\lambda}}(\cdot)$ is considerable compared with the normal one and $\mathbf{prox}_{\frac{1}{2\beta}(\|\cdot\|_k^{sp})^2}(\cdot)$. It is also slightly better than the $\mathbf{prox}_{\frac{\lambda}{2}\|\cdot\|_\Theta^2}(\cdot)$.

### 4.2 Statistical Recovery on Synthetic Data

**Data generation** We fixed $p = 600$, and $\boldsymbol{\theta}^* = (\underbrace{10,\ldots,10}_{10}, \underbrace{10,\ldots,10}_{10}, \underbrace{10,\ldots,10}_{10}, \underbrace{0,0,\ldots,0}_{570})$ throughout the experiment, in which nonzero entries were divided equally into three groups. The design matrix $\mathbf{X}$ were generated from a normal distribution such that the entries in the same group

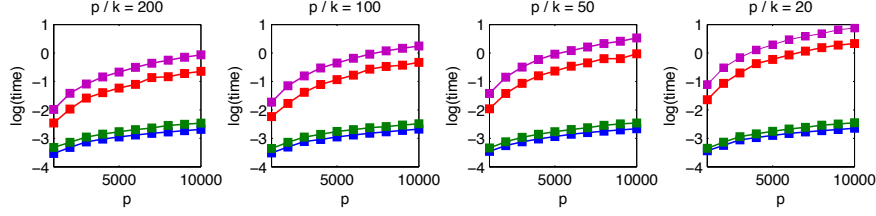

*Figure 1:* Efficiency of proximal operators. Diamond: $\mathbf{prox}_{\mathbb{I}_{\mathcal{C}_\lambda}}(\cdot)$ in Theorem 2, Square: $\mathbf{prox}_{\frac{1}{2\beta}(\|\cdot\|_k^{sp})^2}(\cdot)$ in [1], Downward-pointing triangle: $\mathbf{prox}_{\frac{\lambda}{2}\|\cdot\|_\Theta^2}(\cdot)$ in [13], Upward-pointing triangle: accelerated $\mathbf{prox}_{\mathbb{I}_{\mathcal{C}_\lambda}}(\cdot)$ in Theorem 3. For each $(p, k)$, 200 vectors are randomly generated for testing. Time is measured in seconds.

have the same mean sampled from $\mathcal{N}(0, 1)$. $\mathbf{X}$ was normalized afterwards. The response vector $\mathbf{y}$ was given by $\mathbf{y} = \mathbf{X}\boldsymbol{\theta}^* + 0.01 \times \mathcal{N}(0, 1)$. The number of samples $n$ is specified later.

**ROC curves with different $k$** We fixed $n = 400$ to obtain the ROC plot for $k = \{1, 10, 50\}$ as shown in Figure 2(a). $\lambda_p$ ranged from $10^{-2}$ to $10^3$. Small $k$ gets better ROC curve.

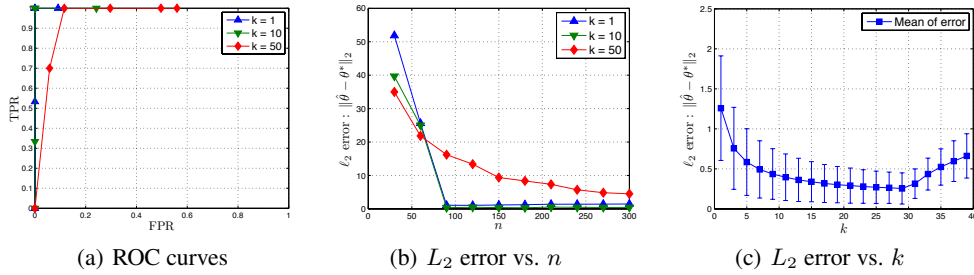

| (a) ROC curves | (b) $L_2$ error vs. $n$ | (c) $L_2$ error vs. $k$ |

*Figure 2:* (a) The true positive rate reaches 1 quite early for $k = 1, 10$. When $k = 50$, the ROC gets worse due to the strong smoothing effect introduced by large $k$. (b) For each $k$, the $L_2$ error is large when the sample is inadequate. As $n$ increases, the error decreases dramatically for $k = 1, 10$ and becomes stable afterwards, while the decrease is not that significant for $k = 50$ and the error remains relatively large. (c) Both mean and standard deviation of $L_2$ error are decreasing as $k$ increases until it exceeds the number of nonzero entries in $\boldsymbol{\theta}^*$, and then the error goes up for larger $k$.

$L_2$ **error vs. $n$** We investigated how the $L_2$ error $\|\hat{\boldsymbol{\theta}} - \boldsymbol{\theta}^*\|_2$ of Dantzig selector changes as the number of samples increases, where $k = \{1, 10, 50\}$ and $n = \{30, 60, 90, \ldots, 300\}$. $k = 1, 10$ give small errors when $n$ is sufficiently large.

$L_2$ **error vs. $k$** We also looked at the $L_2$ error with different $k$. We again fixed $n = 400$ and varied $k$ from 1 to 39. For each $k$, we repeated the experiment 100 times, and obtained the mean and standard deviation plot in Figure 2(c). The result shows that the $k$-support-norm GDS with suitable $k$ outperforms the $L_1$-norm DS (i.e. $k = 1$) when correlated variables present in data.

## 5   Conclusions

In this paper, we introduced the GDS, which generalizes the standard $L_1$-norm Dantzig selector to estimation with any norm, such that structural information encoded in the norm can be efficiently exploited. A flexible framework based on inexact ADMM is proposed for solving the GDS, which only requires one of conjugate proximal operators to be efficiently solved. Further, we provide a unified statistical analysis framework for the GDS, which utilizes Gaussian widths of certain restricted sets for proving consistency. In the non-trivial example of $k$-support norm, we showed that the proximal operators used in the inexact ADMM can be computed more efficiently compared to previously proposed variants. Our statistical analysis for the $k$-support norm provides the first result of consistency of this structured norm. Further, experimental results provided sound support to the theoretical development in the paper.

### Acknowledgements

The research was supported by NSF grants IIS-1447566, IIS-1422557, CCF-1451986, CNS-1314560, IIS-0953274, IIS-1029711, and by NASA grant NNX12AQ39A.

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
