[Supplementary Material]

# Supplementary Material to Generalized Dantzig Selector: Application to the $k$-support norm

## 1 Proof of Theorem 1

**Statement of Theorem:** *Suppose that both design matrix $\mathbf{X}$ and noise $\mathbf{w}$ consists of i.i.d. Gaussian entries with zero mean variance 1 and $\mathbf{X}$ has normalized columns, i.e. $\|\mathbf{X}^{(j)}\|_2 = 1$, $j = 1, \ldots, p$. If we solve the problem (1) with*

$$\lambda_p \geq c\mathbf{E}\left[\mathcal{R}^*(\mathbf{X}^T\mathbf{w})\right] , \tag{A.1}$$

*where $c > 1$ is a constant, then, with probability at least $(1 - \eta_1 \exp(-\eta_2 n))$, we have*

$$\|\hat{\boldsymbol{\theta}} - \boldsymbol{\theta}^*\|_2 \leq \frac{4\sqrt{\mathcal{R}(\boldsymbol{\theta}^*)\lambda_p}}{(\ell_n - \omega(\mathcal{T}_{\mathcal{R}}(\boldsymbol{\theta}^*) \cap \mathbb{S}^{p-1}))} , \tag{A.2}$$

*where $\omega(\mathcal{T}_{\mathcal{R}}(\boldsymbol{\theta}^*) \cap \mathbb{S}^{p-1})$ is the Gaussian width of the intersection of the error cone $\mathcal{T}_{\mathcal{R}}(\boldsymbol{\theta}^*)$ and the unit spherical shell $\mathbb{S}^{p-1}$, and $\ell_n$ is the expected length of a length $n$ i.i.d. standard Gaussian vector with $\frac{n}{\sqrt{n+1}} < \ell_n < \sqrt{n}$, and $\eta_1, \eta_2 > 0$ are constants.*

*Proof:* We use the following lemma for the proof.

**Lemma 1** *Suppose we solve the minimization problem (1) with $\lambda_p \geq \mathcal{R}^*\left(\mathbf{X}^T\mathbf{w}\right)$. Then the error vector $\hat{\Delta}$ belongs to the set*

$$\mathcal{T}_{\mathcal{R}}(\boldsymbol{\theta}^*) := \text{cone}\left\{\Delta \in \mathbb{R}^p \ : \ \mathcal{R}(\boldsymbol{\theta}^* + \Delta) \leq \mathcal{R}(\boldsymbol{\theta}^*)\right\} , \tag{A.3}$$

*and the error $\hat{\Delta} = \hat{\boldsymbol{\theta}} - \boldsymbol{\theta}^*$ satisfies the following bound*

$$\mathcal{R}^*\left(\mathbf{X}^T\mathbf{X}\hat{\Delta}\right) \leq 2\lambda_p \tag{A.4}$$

*Proof:* By our choice of $\lambda_p$, both $\boldsymbol{\theta}^*$ and $\hat{\boldsymbol{\theta}}$ lie in the feasible set of (1), and by optimality of $\hat{\boldsymbol{\theta}}$,

$$\mathcal{R}\left(\boldsymbol{\theta}^* + \hat{\Delta}\right) = \mathcal{R}(\hat{\boldsymbol{\theta}}) \leq \mathcal{R}(\boldsymbol{\theta}^*) . \tag{A.5}$$

Also, by triangle inequality

$$\mathcal{R}^*\left(\mathbf{X}^T\mathbf{X}\hat{\Delta}\right) = \mathcal{R}^*\left(\mathbf{X}^T\mathbf{X}(\hat{\boldsymbol{\theta}} - \boldsymbol{\theta}^*)\right) \tag{A.6}$$

$$\leq \mathcal{R}^*\left(\mathbf{X}^T(\mathbf{y} - \mathbf{X}\boldsymbol{\theta}^*)\right) + \mathcal{R}^*\left(\mathbf{X}^T(\mathbf{y} - \mathbf{X}\hat{\boldsymbol{\theta}})\right) \leq 2\lambda_p . \tag{A.7}$$

∎

Now, note that since $R^*(\cdot)$ is Lipschitz continuous, choosing $\lambda_p \geq c\mathbf{E}\left[\mathcal{R}^*(\mathbf{X}^T\mathbf{w})\right]$ ensures that $\lambda_p \geq \mathcal{R}^*(\mathbf{X}^T\mathbf{w})$ with high probability, by Gaussian concentration on Lipschitz functions [2]. Then,

both $\boldsymbol{\theta}^*$ and $\hat{\boldsymbol{\theta}}$ lie in the feasible set of (1), since $\mathcal{R}^* \left( \mathbf{X}^T(\mathbf{y} - \mathbf{X}\boldsymbol{\theta}^*) \right) = \mathcal{R}^* \left( \mathbf{X}^T \mathbf{w} \right) \leq \lambda_p$ by the choice of $\lambda_p$. Also, from Lemma 1, we have

$$\mathcal{R}^* \left( \mathbf{X}^T \mathbf{X} \hat{\Delta} \right) \leq 2\lambda_p \tag{A.8}$$

Now, note that

$$\|\mathbf{X}\hat{\Delta}\|_2^2 = \langle \hat{\Delta}, \mathbf{X}^T \mathbf{X} \hat{\Delta} \rangle \leq |\langle \hat{\Delta}, \mathbf{X}^T \mathbf{X} \hat{\Delta} \rangle| \leq \mathcal{R}(\hat{\Delta}) \mathcal{R}^* \left( \mathbf{X}^T \mathbf{X} \hat{\Delta} \right) \leq 2\lambda_p \mathcal{R}(\hat{\Delta}) \,, \tag{A.9}$$

where we have used Holder's inequality, and the bound $\mathcal{R}^* \left( \mathbf{X}^T \mathbf{X} \hat{\Delta} \right) \leq 2\lambda_p$ from above.

Next, we use the definition of the error set (A.3) and triangle inequality to obtain

$$\mathcal{R}(\hat{\Delta}) - \mathcal{R}(\boldsymbol{\theta}^*) \leq \mathcal{R}(\boldsymbol{\theta}^* + \hat{\Delta}) \leq \mathcal{R}(\boldsymbol{\theta}^*) \,, \tag{A.10}$$

so that

$$\mathcal{R}(\hat{\Delta}) \leq 2\mathcal{R}(\boldsymbol{\theta}^*) \,, \tag{A.11}$$

and we obtain the bound

$$\|\mathbf{X}\hat{\Delta}\|_2^2 \leq 4\lambda_p \mathcal{R}(\boldsymbol{\theta}^*) \,. \tag{A.12}$$

Lastly, we use Gordon's theorem, which states that for $\mathbf{X}$ with i.i.d. Gaussian $(0, 1)$ entries,

$$\mathbf{E} \left[ \min_{\mathbf{z} \in \mathcal{T}_\mathcal{R}(\boldsymbol{\theta}^*) \cap \mathbb{S}^{p-1}} \|\mathbf{X}\mathbf{z}\|_2 \right] \geq \ell_n - \omega \left( \mathcal{T}_\mathcal{R}(\boldsymbol{\theta}^*) \cap \mathbb{S}^{p-1} \right) \,, \tag{A.13}$$

where $\ell_n$ is the expected length of an i.i.d. Gaussian random vector of length $n$, and $\omega \left( \mathcal{T}_\mathcal{R}(\boldsymbol{\theta}^*) \cap \mathbb{S}^{p-1} \right)$ is the Gaussian width of the set $\Omega = \left( \mathcal{T}_\mathcal{R}(\boldsymbol{\theta}^*) \cap \mathbb{S}^{p-1} \right)$. Now, since the function $\mathbf{X} \to \min_{\mathbf{z} \in \Omega} \|\mathbf{X}\mathbf{z}\|_2$ is Lipschitz continuous with constant 1 over the set $\Omega$, we can use Gaussian concentration of Lipschitz functions [2] to obtain

$$\|\mathbf{X}\hat{\Delta}\|_2 \geq \frac{1}{2} \left( \ell_n - \omega(\mathcal{T}_\mathcal{R}(\boldsymbol{\theta}^*) \cap \mathbb{S}^{p-1}) \right) \|\hat{\Delta}\|_2 \tag{A.14}$$

with probability greater than $1 - \exp\left( -\frac{1}{8} \left( \ell_n - \omega(\mathcal{T}_\mathcal{R}(\boldsymbol{\theta}^*) \cap \mathbb{S}^{p-1}) \right)^2 \right)$, where $c_1, c_2 > 0$ are constants . Combining (A.14) and (A.12), we obtain

$$\|\hat{\Delta}\|_2 \leq \frac{4\sqrt{\mathcal{R}(\boldsymbol{\theta}^*)\lambda_p}}{(\ell_n - \omega(\mathcal{T}_\mathcal{R}(\boldsymbol{\theta}^*) \cap \mathbb{S}^{p-1}))} \tag{A.15}$$

with probability greater than $1 - \exp\left( -\frac{1}{8} \left( \ell_n - \omega(\mathcal{T}_\mathcal{R}(\boldsymbol{\theta}^*) \cap \mathbb{S}^{p-1}) \right)^2 \right)$, and the statement of the theorem follows.

∎

## 2 Proof of Theorem 2

Given a vector $\mathbf{x}$, we use the notation $\mathbf{x}_{i:j}$ to denote its subvector $(\mathbf{x}_i, \mathbf{x}_{i+1}, \ldots, \mathbf{x}_j)$.

**Statement of Theorem:** *Given $\lambda > 0$ and $\mathbf{x} \in \mathbb{R}^p$, if $\|\mathbf{x}\|_k^{sp^*} \leq \lambda$, then $\mathbf{w}^* = \mathbf{prox}_{\mathbb{I}_{C_\lambda}}(\mathbf{x}) = \mathbf{x}$. If $\|\mathbf{x}\|_k^{sp^*} > \lambda$, define $A_{sr} = \sum_{i=s+1}^r |\mathbf{x}|_i^\downarrow$, $B_s = \sum_{i=1}^s (|\mathbf{x}|_i^\downarrow)^2$, in which $0 \leq s < k$ and $k \leq r \leq p$, and construct the nonlinear equation of $\beta$,*

$$(k - s)A_{sr}^2 \left[ \frac{1 + \beta}{r - s + (k - s)\beta} \right]^2 - \lambda^2(1 + \beta)^2 + B_s = 0 \,. \tag{A.16}$$

*Let $\beta_{sr}$ be given by*

$$\beta_{sr} = \begin{cases} \textit{nonnegative root of (A.16)} & \textit{if } s > 0 \textit{ and the root exists} \\ 0 & \textit{otherwise} \end{cases} \,. \tag{A.17}$$

*Then the proximal operator $\mathbf{w}^* = \mathbf{prox}_{\mathbb{I}_{\mathcal{C}_\lambda}}(\mathbf{x})$ is given by*

$$|\mathbf{w}^*|_i^\downarrow = \begin{cases} \frac{1}{1+\beta_{s^*r^*}}|\mathbf{x}|_i^\downarrow & \text{if } 1 \le i \le s^* \\ \sqrt{\frac{\lambda^2 - B_{s^*}}{k-s^*}} & \text{if } s^* < i \le r^* \text{ and } \beta_{s^*r^*} = 0 \\ \frac{A_{s^*r^*}}{r^* - s^* + (k-s^*)\beta_{s^*r^*}} & \text{if } s^* < i \le r^* \text{ and } \beta_{s^*r^*} > 0 \\ |\mathbf{x}|_i^\downarrow & \text{if } r^* < i \le p \end{cases}, \tag{A.18}$$

*where the indices $s^*$ and $r^*$ with computed $|\mathbf{w}^*|^\downarrow$ make the following two inequalities hold,*

$$|\mathbf{w}^*|_{s^*}^\downarrow > |\mathbf{w}^*|_k^\downarrow, \tag{A.19}$$

$$|\mathbf{x}|_{r^*+1}^\downarrow \le |\mathbf{w}^*|_k^\downarrow < |\mathbf{x}|_{r^*}^\downarrow. \tag{A.20}$$

*There might be multiple pairs of $(s, r)$ satisfying the inequalities (A.19)-(A.20), and we choose the pair with the smallest $\||\mathbf{x}|^\downarrow - |\mathbf{w}|^\downarrow\|_2$. Finally, $\mathbf{w}^*$ is obtained by sign-changing and reordering $|\mathbf{w}^*|^\downarrow$ to conform to $\mathbf{x}$.*

*Proof:* Let $\mathbf{w}^* = \mathbf{prox}_{\mathbb{I}_{\mathcal{C}_\lambda}}(\mathbf{x}) = \operatorname{argmin}_{\mathbf{w} \in \mathcal{C}_\lambda} \frac{1}{2}\|\mathbf{x} - \mathbf{w}\|_2^2$. For simplicity, we drop the constant $\frac{1}{2}$ in later discussion. We consider the following two cases.

**Case 1:** if $\|\mathbf{x}\|_k^{sp^*} \le \lambda$, it is trivial that $\mathbf{w}^* = \mathbf{x}$, which is also the global minimizer of $\|\mathbf{x} - \mathbf{w}\|_2^2$ without the constraint $\mathbf{x} \in \mathcal{C}_\lambda$.

**Case 2:** if $\|\mathbf{x}\|_k^{sp^*} > \lambda$, first we start by noting that, given $\mathbf{x}$ and $\mathbf{w}$, $\|\mathbf{x} - \mathbf{w}\|_2^2 = \|\mathbf{x}\|_2^2 - 2\langle \mathbf{x}, \mathbf{w}\rangle + \|\mathbf{w}\|_2^2 \ge \|\mathbf{x}\|_2^2 - 2\langle|\mathbf{x}|^\downarrow, |\mathbf{w}|^\downarrow\rangle + \|\mathbf{w}\|_2^2$, which implies that $\mathbf{w}^*$ should achieve this lower bound by conforming with the signs and orders of elements in $\mathbf{x}$. Without loss of generality, we are simply focused on the case where $\mathbf{x} = |\mathbf{x}|^\downarrow$.

For $\mathbf{w}^*$ to be the optimal, $\mathbf{w}_{k:p}^*$ should be chosen such that $\mathbf{w}_{k:r}^* = (\mathbf{w}_k^*, \mathbf{w}_k^*, \ldots, \mathbf{w}_k^*)$ and $\mathbf{w}_{r+1:p}^* = \mathbf{x}_{r+1:p}$, where $r$ satisfies $\mathbf{x}_r > \mathbf{w}_k^* \ge \mathbf{x}_{r+1}$, otherwise either the decreasing order of $\mathbf{w}^*$ will be violated or the $\|\mathbf{x}_{k:p} - \mathbf{w}_{k:p}\|_2$ is not minimized. As for $\mathbf{w}_{1:k-1}^*$, we similarly assume $\mathbf{w}_{s+1:k-1}^* = (\mathbf{w}_k^*, \mathbf{w}_k^*, \ldots, \mathbf{w}_k^*)$ for some $0 \le s \le k-1$, then $\mathbf{w}_{1:s}^*$ should be chosen to minimize $\|\mathbf{x}_{1:s} - \mathbf{w}_{1:s}\|_2$ such that $\|\mathbf{w}_{1:s}\|_2^2 = \|\mathbf{w}_{1:k}^*\|_2^2 - \|\mathbf{w}_{s+1:k}^*\|_2^2 \le \lambda^2 - (k-s)(\mathbf{w}_k^*)^2$. By Cauchy-Schwarz Inequality, we note that

$$\|\mathbf{x}_{1:s} - \mathbf{w}_{1:s}\|_2^2 = \|\mathbf{x}_{1:s}\|_2^2 - 2\langle\mathbf{x}_{1:s}, \mathbf{w}_{1:s}\rangle + \|\mathbf{w}_{1:s}\|_2^2$$
$$\ge \|\mathbf{x}_{1:s}\|_2^2 - 2\|\mathbf{x}_{1:s}\|_2\|\mathbf{w}_{1:s}\|_2 + \|\mathbf{w}_{1:s}\|_2^2$$

where the equality holds when $\mathbf{w}_{1:s}^*$ follows the form of $\mathbf{w}_{1:s}^* = \frac{1}{1+\beta_{sr}}\mathbf{x}_{1:s}$, and $\beta_{sr} \ge 0$ satisfies the constraint $\frac{B_s}{(1+\beta_{sr})^2} = \lambda^2 - (k-s)(\mathbf{w}_k)^2$.

So far we have figured out the structure of $\mathbf{w}^* = (\mathbf{w}_{1:s}^*, \mathbf{w}_{s+1:r}^*, \mathbf{w}_{r+1:p}^*)$, in which the three subvectors, compared with $\mathbf{x}$, are shrunk by a common factor $1 + \beta_{sr}$, constant $\mathbf{w}_k^*$, or unchanged. Next we need to determine the value of $\beta_{sr}$ and $\mathbf{w}_k^*$. By optimality, $\|\mathbf{x} - \mathbf{w}\|_2^2 = \|\mathbf{x}_{1:r} - \mathbf{w}_{1:r}\|_2^2$ must be minimized at $\mathbf{w}^*$, so we have the following problem,

$$\min_{\beta, \mathbf{w}_k} \|\mathbf{x}_{1:r} - \mathbf{w}_{1:r}\|_2^2 = \|\mathbf{x}_{1:s} - \mathbf{w}_{1:s}\|_2^2 + \|\mathbf{x}_{s+1:r} - \mathbf{w}_{s+1:r}\|_2^2$$
$$= (\frac{\beta}{1+\beta})^2 B_s + \sum_{i=s+1}^{r}(\mathbf{x}_i - \mathbf{w}_k)^2 \tag{A.21}$$

$$\text{s.t.} \quad (\|\mathbf{w}\|_k^{sp^*})^2 = \frac{B_s}{(1+\beta)^2} + (k-s)(\mathbf{w}_k)^2 = \lambda^2 \tag{A.22}$$

Replacing $\mathbf{w}_k$ in (A.21) with $\mathbf{w}_k = \sqrt{\frac{\lambda^2 - \frac{B_s}{(1+\beta)^2}}{k-s}}$ obtained from (A.22), we express $\|\mathbf{x}_{1:r} - \mathbf{w}_{1:r}\|_2^2$ as a function of $\beta$,

$$\Phi_{sr}(\beta) = (\frac{\beta}{1+\beta})^2 B_s + \sum_{i=s+1}^{r}\left(\mathbf{x}_i - \sqrt{\frac{\lambda^2 - \frac{B_s}{(1+\beta)^2}}{k-s}}\right)^2 \tag{A.23}$$

Set derivative of $\Phi_{sr}(\beta)$ to be zero, we have

$$\frac{d}{d\beta}\Phi_{sr}(\beta) = \frac{d}{d\beta}\Big[(\frac{\beta}{1+\beta})^2 B_s + \sum_{i=s+1}^{r}\big(\mathbf{x}_i - \sqrt{\frac{\lambda^2 - \frac{B_s}{(1+\beta)^2}}{k-s}}\big)^2\Big] \tag{A.24}$$

$$= \frac{2\beta}{(1+\beta)^3}B_s - \frac{2A_{sr}B_s}{(1+\beta)^3(k-s)\sqrt{\frac{\lambda^2 - \frac{B_s}{(1+\beta)^2}}{k-s}}} + \frac{2(r-s)B_s}{(k-s)(1+\beta)^3} \tag{A.25}$$

$$= \frac{2B_s}{(k-s)(1+\beta)^3}\Big[(k-s)\beta - \frac{A_{sr}}{\sqrt{\frac{\lambda^2 - \frac{B_s}{(1+\beta)^2}}{k-s}}} + (r-s)\Big] = 0 \tag{A.26}$$

If $s > 0$, then $B_s > 0$ and (A.26) is equivalent to (A.16). And we can see that the quantity inside the bracket of (A.26) is monotonically increasing when $\beta \geq \max(0, \frac{\sqrt{B_s}-\lambda}{\lambda})$, thus ensuring the nonnegative root $\beta_{sr}$ is unique if existing. If the nonnegative root exists, the expression for $\mathbf{w}^*_{s+1:r}$ can be obtained from (A.26), whose entries are all equal to $\mathbf{w}^*_k$.

If $s > 0$ and a nonnegative root of (A.26) is nonexistent, the derivative is always positive when $\beta \geq 0$, which means that $\Phi_{sr}(\beta)$ is increasing. Hence the minimizer of $\Phi_{sr}(\beta)$ is $\beta_{sr} = 0$. If $s = 0$, we actually do not care about the value of $\beta_{sr}$ because the problem defined by (A.21) and (A.22) is independent of $\beta$, and we set it to be 0 for simplicity. According to (A.22), both cases of $\beta_{sr} = 0$ lead to the same expression for $\mathbf{w}^*_{s+1:r}$ in (A.18).

As we do not know beforehand which $s$ and $r$ to choose, we need to search for $r^*$ and $s^*$ that gives the smallest $\||\mathbf{x}|^{\downarrow} - |\mathbf{w}|^{\downarrow}\|_2$, and also to check whether the $\mathbf{w}^*$ obtained by (A.18) is in decreasing order, which are the conditions (A.19) and (A.20) presented in Theorem 2. ∎

## 3 Proof of Theorem 3

**Statement of Theorem:** *In search of $(s^*, r^*)$ defined in Theorem 2, there can be only one $\tilde{r}$ for a given candidate $\tilde{s}$ of $s^*$, such that the inequality (A.20) is satisfied. Moreover if such $\tilde{r}$ exists, then for any $r < \tilde{r}$, the associated $|\tilde{\mathbf{w}}|^{\downarrow}_k$ violates the first part of (A.20), and for $r > \tilde{r}$, $|\tilde{\mathbf{w}}|^{\downarrow}_k$ violates the second part of (A.20). On the other hand, based on the $\tilde{r}$, we have following assertion of $s^*$,*

$$s^* \begin{cases} > \tilde{s} & \text{if } \tilde{r} \text{ does not exist} \\ \geq \tilde{s} & \text{if } \tilde{r} \text{ exists and corresponding } |\tilde{\mathbf{w}}|^{\downarrow}_k \text{ satisfies (A.19)} \\ < \tilde{s} & \text{if } \tilde{r} \text{ exists but corresponding } |\tilde{\mathbf{w}}|^{\downarrow}_k \text{ violates (A.19)} \end{cases} \tag{A.27}$$

To prove Theorem 3, we first need the following corollary from Theorem 2.

**Corollary 1** *When $\beta \geq \max(0, \frac{\sqrt{B_s}-\lambda}{\lambda})$, $\Phi_{sr}(\beta)$ defined in (A.23) is decreasing when $\beta < \beta_{sr}$, and increasing when $\beta > \beta_{sr}$. Equivalently, $\Phi_{sr}(\beta) = \|\mathbf{x}_{1:r} - \mathbf{w}_{1:r}\|_2^2$, when treated as function of $\mathbf{w}_k$, is decreasing when $\mathbf{w}_k < \mathbf{w}^*_k$ and increasing when $\mathbf{w}_k > \mathbf{w}^*_k$.*

*Proof:* The first part simply follows the monotonicity of $\frac{d}{d\beta}\Phi_{sr}(\beta)$ mentioned in the proof of Theorem 2, which implies that $\frac{d}{d\beta}\Phi_{sr}(\beta)$ is negative when $\beta < \beta_{sr}$, and positive when $\beta > \beta_{sr}$. The constraint (A.22) implies that $\mathbf{w}_k$ increases as $\beta$ increases. So $\|\mathbf{x}_{1:r} - \mathbf{w}_{1:r}\|_2^2$, as a function of $\mathbf{w}_k$, has the same monotonicity w.r.t. $\mathbf{w}_k$. ∎

Now we present the proof of Theorem 3.

*Proof:* First we show by contradiction that for a given $s$, the $\tilde{r}$ that satisfies (A.20) can be at most one. Suppose there are two indices, say $r_1$ and $r_2$, which satisfy that condition with a certain $s$.

Without loss of generality, let $r_1 < r_2$, we know that their corresponding $\mathbf{w}^{(1)}$ and $\mathbf{w}^{(2)}$ should minimize $\|\mathbf{x}_{1:r_1} - \mathbf{w}_{1:r_1}\|_2^2$ and $\|\mathbf{x}_{1:r_2} - \mathbf{w}_{1:r_2}\|_2^2$, respectively. As $r_1 < r_2$, then $\mathbf{w}_k^{(1)} \geq \mathbf{x}_{r_2} > \mathbf{w}_k^{(2)}$ according to (A.20). Construct

$$\mathbf{w}' = (\underbrace{\frac{\mathbf{x}_1}{1+\beta'}, \ldots, \frac{\mathbf{x}_s}{1+\beta'}}_{s}, \underbrace{\mathbf{x}_{r_2}, \ldots, \mathbf{x}_{r_2}}_{r_2 - s}, \mathbf{x}_{r_2+1}, \ldots, \mathbf{x}_p)$$

where $\beta'$ is chosen to satisfy the constraint (A.22) with $\mathbf{w}_k' = \mathbf{x}_{r_2}$, and $\|\mathbf{x}_{1:r_2} - \mathbf{w}_{1:r_2}^{(2)}\|_2^2$ can be decomposed as

$$\|\mathbf{x}_{1:r_2} - \mathbf{w}_{1:r_2}^{(2)}\|_2^2 = \|\mathbf{x}_{1:r_1} - \mathbf{w}_{1:r_1}^{(2)}\|_2^2 + \|\mathbf{x}_{r_1+1:r_2} - \mathbf{w}_{r_1+1:r_2}^{(2)}\|_2^2$$
$$> \|\mathbf{x}_{1:r_1} - \mathbf{w}_{1:r_1}'\|_2^2 + \|\mathbf{x}_{r_1+1:r_2} - \mathbf{w}_{r_1+1:r_2}'\|_2^2$$
$$= \|\mathbf{x}_{1:r_2} - \mathbf{w}_{1:r_2}'\|_2^2$$

which contradicts that $\mathbf{w}_{1:r_2}^{(2)}$ minimize $\|\mathbf{x}_{1:r_2} - \mathbf{w}_{1:r_2}\|_2^2$. Note that $\|\mathbf{x}_{1:r_1} - \mathbf{w}_{1:r_1}^{(2)}\|_2^2 > \|\mathbf{x}_{1:r_1} - \mathbf{w}_{1:r_1}'\|_2^2$ simply follows Corollary 1 as $\mathbf{w}_k^{(1)} \geq \mathbf{x}_{r_2} = \mathbf{w}_k' > \mathbf{w}_k^{(2)}$, and $\|\mathbf{x}_{r_1+1:r_2} - \mathbf{w}_{r_1+1:r_2}^{(2)}\|_2^2 > \|\mathbf{x}_{r_1+1:r_2} - \mathbf{w}_{r_1+1:r_2}'\|_2^2$ is due to the fact that $\mathbf{x}_{r_1+1} \geq \ldots \geq \mathbf{x}_{r_2} = \mathbf{w}_k' > \mathbf{w}_k^{(2)}$.

Next we show by contradiction that if $\tilde{r}$ exists for given $s$, then any $r < \tilde{r}$ violates the first part of (A.20), and any $r > \tilde{r}$ violates second part. Let $\tilde{\mathbf{w}}$ denote the minimizer of $\|\mathbf{x}_{1:\tilde{r}} - \mathbf{w}_{1:\tilde{r}}\|_2^2$. Suppose $r < \tilde{r}$ and the first part of (A.20) is not violated, then its second part must be violated due to the uniqueness of $\tilde{r}$. Then we can construct new

$$\mathbf{w}' = (\underbrace{\frac{\mathbf{x}_1}{1+\beta'}, \ldots, \frac{\mathbf{x}_s}{1+\beta'}}_{s}, \underbrace{\mathbf{x}_{\tilde{r}}, \ldots, \mathbf{x}_{\tilde{r}}}_{\tilde{r} - s}, \mathbf{x}_{\tilde{r}+1}, \ldots, \mathbf{x}_p),$$

where $\beta'$ is again chosen to satisfy the constraint (A.22) with $\mathbf{w}_k' = \mathbf{x}_{\tilde{r}}$. This by the same argument for proving the uniqueness of $\tilde{r}$ make the following inequality hold,

$$\|\mathbf{x}_{1:\tilde{r}} - \tilde{\mathbf{w}}_{1:\tilde{r}}\|_2^2 = \|\mathbf{x}_{1:r} - \tilde{\mathbf{w}}_{1:r}\|_2^2 + \|\mathbf{x}_{r+1:\tilde{r}} - \tilde{\mathbf{w}}_{r+1:\tilde{r}}\|_2^2$$
$$> \|\mathbf{x}_{1:r} - \mathbf{w}_{1:r}'\|_2^2 + \|\mathbf{x}_{r+1:\tilde{r}} - \mathbf{w}_{r+1:\tilde{r}}'\|_2^2$$
$$= \|\mathbf{x}_{1:\tilde{r}} - \mathbf{w}_{1:\tilde{r}}'\|_2^2 .$$

This contradicts that $\tilde{\mathbf{w}}$ is the minimizer of $\|\mathbf{x}_{1:\tilde{r}} - \mathbf{w}_{1:\tilde{r}}\|_2^2$. Similar argument applies to the case when $r > \tilde{r}$. We construct another

$$\mathbf{w}'' = (\underbrace{\frac{\mathbf{x}_1}{1+\beta''}, \ldots, \frac{\mathbf{x}_s}{1+\beta''}}_{s}, \underbrace{\mathbf{x}_{r+1}, \ldots, \mathbf{x}_{r+1}}_{r-s}, \mathbf{x}_{r+1}, \ldots, \mathbf{x}_p),$$

which gives smaller $\|\mathbf{x}_{1:r} - \mathbf{w}_{1:r}\|_2^2$ than any $\mathbf{w}$ with $\mathbf{w}_k < \mathbf{x}_{r+1}$ according to Corollary 1. Therefore it is impossible for $r > \tilde{r}$ to violate the first inequality. Note that $\beta''$ together with $\mathbf{w}_k'' = \mathbf{x}_{r+1}$ satisfies (A.22).

Finally we show the assertion (A.27) for $s^*$. We note that when $\tilde{s}$ is fixed, finding solution to the proximal operator can be regarded as finding the minimizer of (A.21) under the constraint $\mathbf{w}_k = \mathbf{w}_{k-1} = \ldots = \mathbf{w}_{\tilde{s}+1}$. So for $s < \tilde{s}$, the minimization problem is equivalent to the one for $\tilde{s}$ under additional constraint $\mathbf{w}_{\tilde{s}+1} = \mathbf{w}_{\tilde{s}} = \ldots = \mathbf{w}_{s+1}$. Therefore if $\tilde{r}$ does not exist or $|\tilde{\mathbf{w}}|_k^\downarrow$ already satisfies (A.19), then $s^* \geq \tilde{s}$ because $s < \tilde{s}$ considers a more restricted problem and is unable to get a better result.

For the situation in which $\tilde{r}$ exists for $\tilde{s}$ but associated $|\tilde{\mathbf{w}}|_k^\downarrow$ violates (A.19), we show by contradiction that for any $s' > \tilde{s}$, (A.19) is also violated. Assume that there is a solution $\mathbf{w}'$ satisfying both (A.19) and (A.20) for $s' = \tilde{s} + 1$ and the corresponding $\tilde{r}'$. It is not difficult to see that $|\mathbf{w}'|_k^\downarrow < |\tilde{\mathbf{w}}|_k^\downarrow$ and $\tilde{r}' \geq \tilde{r}$. By the violation we have shown, we know that the minimizer of (A.21) for $(s', \tilde{r})$, denoted by $\mathbf{w}''$, satisfies $|\mathbf{w}''|_k^\downarrow \leq |\mathbf{w}'|_k^\downarrow$ (Note that $\mathbf{w}'$ is the minimizer of (A.21) for $(s', \tilde{r}')$). Combined with $|\mathbf{w}'|_k^\downarrow < |\tilde{\mathbf{w}}|_k^\downarrow$, this indicates by Corollary 1 that $\Phi_{s'\tilde{r}}(\cdot)$ increases on the interval $[|\mathbf{w}''|_k^\downarrow, |\tilde{\mathbf{w}}|_k^\downarrow]$. Then we consider two sequential modifications on $\tilde{\mathbf{w}}$,

1. Replacing the $|\tilde{\mathbf{w}}|^{\downarrow}_{1:s'}$ in $|\tilde{\mathbf{w}}|^{\downarrow}$ with $\frac{\||\tilde{\mathbf{w}}|^{\downarrow}_{1:s'}\|_2}{\||\mathbf{x}|^{\downarrow}_{1:s'}\|_2}|\mathbf{x}|^{\downarrow}_{1:s'}$ ,

2. Shrink $|\tilde{\mathbf{w}}|^{\downarrow}_{s'+1:\tilde{r}}$ and amplify the new $|\tilde{\mathbf{w}}|^{\downarrow}_{1:s'}$ by some factor such that (A.22) still holds for $s'$ and $|\tilde{\mathbf{w}}|^{\downarrow}_{s'+1} = |\tilde{\mathbf{w}}|^{\downarrow}_{s'}$ .

Note that the two modifications both decrease $\|\mathbf{x}_{1:\tilde{r}} - \tilde{\mathbf{w}}_{1:\tilde{r}}\|_2$. Decrease in Modification 1 is the result of Cauchy Schwarz Inequality, and decrease in Modification 2 is due to the monotonicity of $\Phi_{s'\tilde{r}}(\cdot)$ we mentioned afront. The modified $\tilde{\mathbf{w}}$ satisfies $|\tilde{\mathbf{w}}|^{\downarrow}_{\tilde{s}+1} = |\tilde{\mathbf{w}}|^{\downarrow}_{\tilde{s}+2} = \ldots = |\tilde{\mathbf{w}}|^{\downarrow}_k$, thus contradicting that the old $\tilde{\mathbf{w}}$ is the minimizer of (A.21) for $(\tilde{s}, \tilde{r})$. Hence, by induction, we conclude that for any $s' > \tilde{s}$, its solution also violates (A.19).

Assembling the conclusions above, we have (A.27) for $s^*$.

∎

# 4 Proof of Theorem 4

**Statement of Theorem:** *For the $k$-support norm Generalized Dantzig Selection problem* (20)*, we obtain*

$$\mathbf{E}\left[\mathcal{R}^*(\mathbf{X}^T\mathbf{w})\right] \leq k\left(\sqrt{2\log\left(\frac{ep}{k}\right)} + 1\right)^2 \tag{A.28}$$

$$\omega(\mathcal{T}_\mathcal{A}(\boldsymbol{\theta}^*) \cap \mathbb{S}^{p-1})^2 \leq \left(\sqrt{2k\log\left((p - k - \left\lceil\frac{s}{k}\right\rceil + 2)\right)} + \sqrt{k}\right)^2 \cdot \left\lceil\frac{s}{k}\right\rceil + s . \tag{A.29}$$

*Proof:* We first illustrate that the $k$-support norm is an atomic norm, and then prove Theorem 4.

## 4.1 $k$-Support norm as an Atomic Norm

Here we show that $k$-support norm satisfies the definition of atomic norms [1]. Consider $\mathcal{G}_j$ to be the set of all subsets of $\{1, 2, \ldots, p\}$ of size $j$, so that

$$\mathcal{G}^{(k)} = \{\mathcal{G}_j\}^k_{j=1} . \tag{A.30}$$

For every $j$, consider the set

$$\mathcal{A}_j = \left\{\mathbf{w} \ : \ \|(\mathbf{w}_{G_j})\|_2 = 1, \ G_j \in \mathcal{G}_j, \ \mathbf{w}_i = \frac{1}{\sqrt{j}}, \ \forall i \in G_j, \ \mathbf{w}_i = 0, \forall i \notin G_j\right\}, \tag{A.31}$$

corresponding to $\mathcal{G}_j$, and the union of such sets

$$\mathcal{A} = \{\mathcal{A}_j\}_{j \in \{1, \ldots, k\}} . \tag{A.32}$$

Note that since every non-zero element in a vector in $\mathcal{A}_j$ is $\frac{1}{\sqrt{j}}$, such an element cannot be represented as a convex combination of elements of the set $\mathcal{A}_l$, $l < j$, whose non-zero elements are $\frac{1}{\sqrt{l}}$. Therefore none of the elements $\mathbf{w}$ in the set $\mathcal{A}$ lies in the convex hull of the other elements $\mathcal{A} \setminus \{\mathbf{w}\}$. Further, note that

$$\text{conv}(\mathcal{A}) = C_k , \tag{A.33}$$

and the $k$-support norm defines the gauge function of the $\mathcal{A}$. Thus the $k$-support norm is an atomic norm.

## 4.2 The Error set and its Gaussian width

Note that the cardinality of the set $\mathcal{G}^{(k)}$ is

$$M = \binom{p}{k} + \binom{p}{k-1} + \binom{p}{k-2} + \cdots + \binom{p}{1} \tag{A.34}$$

The the error set is given by

$$\mathcal{T}_{\mathcal{A}}(\boldsymbol{\theta}^*) = \text{cone}\{\Delta \in \mathbb{R}^p \ : \ \|\Delta + \boldsymbol{\theta}^*\|_k^{sp} \le \|\boldsymbol{\theta}^*\|_k^{sp}\} \ . \tag{A.35}$$

Note that this set is a cone, and we can define the *normal* cone of this set as

$$\mathcal{N}_{\mathcal{A}}(\boldsymbol{\theta}^*) = \{\mathbf{u} \ : \ \langle \mathbf{u}, \Delta \rangle \le 0, \ \forall \Delta \in \mathcal{T}_{\mathcal{A}}(\boldsymbol{\theta}^*)\} \tag{A.36}$$

$$\tag{A.37}$$

The following proposition, shown in [3], shows that the normal cone can be written in terms of the dual norm of the $k$-support norm.

**Proposition 1** *The normal cone to the tangent cone defined in* (A.35) *can written as*

$$\mathcal{N}_{\mathcal{A}}(\boldsymbol{\theta}^*) = \{\mathbf{u} \ : \ \exists t > 0 \ s.t. \ \langle \mathbf{u}, \boldsymbol{\theta}^* \rangle = t\|\boldsymbol{\theta}^*\|_k^{sp}, \ \|\mathbf{u}\|_k^{sp^*} \le t\} \ . \tag{A.38}$$

*Proof:* We re-write the definition of the normal cone in terms of the estimated parameter $\hat{\boldsymbol{\theta}}$ as

$$\mathcal{N}_{\mathcal{A}}(\boldsymbol{\theta}^*) = \{\mathbf{u} \in \mathbb{R}^p \ : \ \langle \mathbf{u}, \boldsymbol{\theta} - \boldsymbol{\theta}^* \rangle \le 0, \forall \boldsymbol{\theta} - \boldsymbol{\theta}^* \in \mathcal{T}_{\mathcal{A}}(\boldsymbol{\theta}^*)\} \ . \tag{A.39}$$

Note that this means that $\mathbf{u} \in \mathcal{N}_{\mathcal{A}}(\boldsymbol{\theta}^*)$ if and only if

$$\langle \mathbf{u}, \boldsymbol{\theta} - \boldsymbol{\theta}^* \rangle \le 0, \ \ \forall \|\boldsymbol{\theta}\|_k^{sp} \le \|\boldsymbol{\theta}^*\|_k^{sp} \tag{A.40}$$

$$\Rightarrow \langle \mathbf{u}, \boldsymbol{\theta} \rangle \le \langle \mathbf{u}, \boldsymbol{\theta}^* \rangle \ \ \forall \|\boldsymbol{\theta}\|_k^{sp} \le \|\boldsymbol{\theta}^*\|_k^{sp} \ . \tag{A.41}$$

Now, we claim that $\langle \mathbf{u}, \boldsymbol{\theta}^* \rangle \ge 0$ for all such $\mathbf{u}$. This can be shown as follows. Assume the contrary, i.e. there exists a $\hat{\mathbf{u}} \in \mathcal{N}_{\mathcal{A}}(\boldsymbol{\theta}^*)$ such that $\langle \hat{\mathbf{u}}, \boldsymbol{\theta}^* \rangle < 0$. Now, noting that $(-\boldsymbol{\theta}^*) \in \mathcal{T}_{\mathcal{A}}(\boldsymbol{\theta}^*)$, we have

$$\langle \hat{\mathbf{u}}, -\boldsymbol{\theta}^* \rangle = -\langle \hat{\mathbf{u}}, \boldsymbol{\theta}^* \rangle > 0 \ , \tag{A.42}$$

so that $\hat{\mathbf{u}} \notin \mathcal{N}_{\mathcal{A}}(\boldsymbol{\theta}^*)$, which is a contradiction, and the claim follows.

Therefore, we can write

$$\langle \mathbf{u}, \boldsymbol{\theta}^* \rangle = t\|\boldsymbol{\theta}^*\|_k^{sp} \tag{A.43}$$

for some $t \ge 0$. Then, $\mathbf{u} \in \mathcal{N}_{\mathcal{A}}(\boldsymbol{\theta}^*)$ if and only if

$$\exists t \ge 0 \ , \ \langle \mathbf{u}, \boldsymbol{\theta}^* \rangle = t\|\boldsymbol{\theta}^*\|_k^{sp} \ , \ \langle \mathbf{u}, \boldsymbol{\theta} \rangle \le t\|\boldsymbol{\theta}^*\|_k^{sp} \ \forall \|\boldsymbol{\theta}\|_k^{sp} \le \|\boldsymbol{\theta}^*\|_k^{sp} \ . \tag{A.44}$$

Since

$$\langle \mathbf{u}, \boldsymbol{\theta} \rangle \le t\|\boldsymbol{\theta}^*\|_k^{sp}, \ \ \forall \|\boldsymbol{\theta}\|_k^{sp} \le \|\boldsymbol{\theta}^*\|_k^{sp} \ \Rightarrow \ \|\mathbf{u}\|_k^{sp^*} \le t \ , \tag{A.45}$$

the statement follows.

∎

The $k$-support norm can be thought of as a group sparse norm with overlaps, such as been dealt with in [3]. Therefore, we can utilize some of the analysis techniques developed in [3], specialized to the structure of the $k$-support norm. We begin by stating a theorem which enables us to bound the Gaussian width of the error set.

First, we define sets that involve the support set of $\boldsymbol{\theta}^*$. Let us define the set $\mathcal{G}^* \subseteq \mathcal{G}^{(k)}$ to be the set of all groups in $\mathcal{G}^{(k)}$ which overlap with the support of $\boldsymbol{\theta}^*$, i.e.

$$\mathcal{G}^* = \{G \in \mathcal{G}^{(k)} \ : \ G \cap \text{supp}(\boldsymbol{\theta}^*) \ne \emptyset\} \ . \tag{A.46}$$

Let $S$ be the union of all groups in $\mathcal{G}^*$, i.e. $S = \bigcup_{G \in \mathcal{G}^*} G$, and the size of $S$ be $|S| = s$. We are going to use three lemmas in order to prove the above bound. The first lemma, proved in [1], upper bounds the Gaussian width by an expected distance to the normal cone as follows.

**Lemma 2 ([1] Proposition 3.6)** *Let* $\mathbb{C}$ *be any nonempty convex in* $\mathbb{R}^p$, *and* $\mathbf{g} \sim \mathcal{N}(0, I_p)$ *be a random gaussian vector. Then*

$$\omega(\mathbb{C} \cap \mathbb{S}^{p-1}) \le \mathbf{E}_{\mathbf{g}}[dist(\mathbf{g}, \mathbb{C}^*)] \ , \tag{A.47}$$

*where* $\mathbb{C}^*$ *is the polar cone of* $\mathbb{C}$.

Note that $\mathcal{N}_{\mathcal{A}}$ is the polar cone of $\mathcal{T}_{\mathcal{A}}$ by definition. Therefore, using Jensen's inequality, we obtain

$$\omega(\mathcal{T}_{\mathcal{A}} \cap \mathbb{S}^{p-1})^2 \le \mathbf{E}_{\mathbf{g}}^2[\text{dist}(\mathbf{g}, \mathcal{N}_{\mathcal{A}})] \le \mathbf{E}_{\mathbf{g}}[\text{dist}(\mathbf{g}, \mathcal{N}_{\mathcal{A}})^2] \le \mathbf{E}_{\mathbf{g}}[\|\mathbf{g} - \mathbf{z}(\mathbf{g})\|_2^2] \ , \tag{A.48}$$

where $\mathbf{z}(\mathbf{g}) \in \mathcal{N}_{\mathcal{A}}$ is a (random) vector constructed to lie always in the normal cone. The construction proceeds as follows.

**Constructing $\mathbf{z}(\mathbf{g})$:**

Note that $\boldsymbol{\theta}^*_{S^c} = 0$. Let us choose a vector $\mathbf{v} \in \mathcal{N}_{\mathcal{A}}$ such that

$$\|\mathbf{v}\|_k^{sp^*} = 1 \text{ and } \mathbf{v}_{S^c} = 0 . \tag{A.49}$$

We can choose an appropriately scaled $\mathbf{v}$ so that

$$\langle \mathbf{v}, \boldsymbol{\theta}^* \rangle = \|\boldsymbol{\theta}^*\|_k^{sp} , \tag{A.50}$$

and let us write without loss of generality $\mathbf{v} = [\mathbf{v}_S \ \mathbf{v}_{S^c}]$.

Next, let $\mathbf{g} \sim \mathcal{N}(0, I_p)$, and write $\mathbf{g} = [\mathbf{g}_S \ \mathbf{g}_{S^c}]$. We define the quantity

$$t(\mathbf{g}) = \max\left\{ \|\mathbf{g}_G\|_2 \ : \ G \in \mathcal{G}^{(k)}, G \subseteq S^c \right\} = \max\left\{ \left( \sum_{i \in G} \mathbf{g}_i^2 \right)^{\frac{1}{2}} \ : \ G \in \mathcal{G}^{(k)}, G \subseteq S^c \right\} , \tag{A.51}$$

and let $\mathbf{z} = \mathbf{z}(\mathbf{g}) = [\mathbf{z}_S \ \mathbf{z}_{S^c}]$ such that

$$\mathbf{z}_S = t(\mathbf{g})\mathbf{v}_S, \quad \mathbf{z}_{S^c} = \mathbf{g}_{S^c} . \tag{A.52}$$

Note that

$$\langle \mathbf{z}, \boldsymbol{\theta}^* \rangle = t(\mathbf{g})\langle \mathbf{v}_S, \boldsymbol{\theta}^*_S \rangle = t(\mathbf{g})\|\boldsymbol{\theta}^*\|_k^{sp} , \tag{A.53}$$

and

$$\|\mathbf{z}\|_k^{sp^*} = \max\left\{ \|\mathbf{z}_G\|_2 \ : \ G \in \mathcal{G}^{(k)} \right\} \tag{A.54}$$

$$= \max\left\{ \max\{\|\mathbf{z}_G\|_2 \ : \ G \in \mathcal{G}^{(k)}, G \subseteq S\} , \ \max\{\|\mathbf{z}_G\|_2 \ : \ G \in \mathcal{G}^{(k)}, G \subseteq S^c\} \right\} \tag{A.55}$$

$$\overset{(a)}{=} \max\left\{ t(\mathbf{g})\|\mathbf{v}\|_k^{sp^*} , \ t(\mathbf{g}) \right\} \tag{A.56}$$

$$= t(\mathbf{g}) \tag{A.57}$$

where $(a)$ follows from the definition of $t(\mathbf{g})$ and the fact that

$$\max\{\|\mathbf{z}_G\|_2 \ : \ G \in \mathcal{G}^{(k)}, G \subseteq S\} = t(\mathbf{g}) \max\{\|\mathbf{v}_G\|_2 \ : \ G \in \mathcal{G}^{(k)}, G \subseteq S\} = t(\mathbf{g})\|\mathbf{v}\|_k^{sp^*} , \tag{A.58}$$

and since $\|\mathbf{v}\|_k^{sp^*} = 1$. Therefore, $\mathbf{z}(\mathbf{g}) \in \mathcal{N}_{\mathcal{A}}(\boldsymbol{\theta}^*)$ by definition in (A.38) .

In order to upper bound the expectation of $t(\mathbf{g})$, we use the following comparison inequality from [3].

**Lemma 3 ([3] Lemma 3.2)** *Let $q_1, q_2, \ldots, q_L$ be $L$, $\chi$-squared random variables with $d$ degrees of freedom. Then*

$$\mathbf{E}\left[ \max_{1 \leq i \leq L} q_i \right] \leq \left( \sqrt{2 \log L} + \sqrt{d} \right)^2 . \tag{A.59}$$

Last, we prove an upper bound on the expected value of $t(\mathbf{g})$, as shown in the following lemma.

**Lemma 4** *Consider $\mathcal{G}^* \subseteq \mathcal{G}^{(k)}$ to be the set of groups intersecting with the support of $\boldsymbol{\theta}^*$, and let $S$ be the union of groups in $\mathcal{G}^*$, such that $s = |S|$. Then,*

$$\mathbf{E}_{\mathbf{g}}[t(\mathbf{g})^2] \leq \left( \sqrt{2k \log \left( p - k - \left\lceil \frac{s}{k} \right\rceil + 2 \right)} + \sqrt{k} \right)^2 . \tag{A.60}$$

*Proof:* Note that

$$\mathbf{E}_{\mathbf{g}}[t(\mathbf{g})^2] = \mathbf{E}_{\mathbf{g}}\left[ \left( \max\left\{ \|\mathbf{g}_G\|_2 \ : \ G \in \mathcal{G}^{(k)}, G \subseteq S^c \right\} \right)^2 \right] \tag{A.61}$$

$$\leq \mathbf{E}_{\mathbf{g}}\left[ \max\left\{ \|\mathbf{g}_G\|_2^2 \ : \ G \in \mathcal{G}^{(k)}, G \subseteq S^c \right\} \right] \tag{A.62}$$

Each term $\|\mathbf{g}_G\|_2^2$ is a $\chi$-squared variable with at most $k$ degrees of freedom. Since the set $S$ has size $s$, the set $\mathcal{G}^*$ has to contain at least $s_k = \left\lceil \frac{s}{k} \right\rceil$ groups of size $k$. Therefore,

$$s = |S| \geq k + (s_k - 1) , \tag{A.63}$$

and therefore the size of its complement is upper bounded by

$$|S^c| \leq p - k - s_k + 1 . \tag{A.64}$$

Therefore the following inequality provides an upper bound on the number of groups involved in computing the maximum in (A.62)

$$\left| \left\{ G \in \mathcal{G}^{(k)}, G \subseteq S^c \right\} \right| \leq \binom{p-k-s_k+1}{k} + \binom{p-k-s_k+1}{k-1} + \cdots + \binom{p-k-s_k+1}{1} \tag{A.65}$$

$$\leq (p - k - s_k + 2)^k \tag{A.66}$$

where we have used the following inequality

$$\binom{n}{h} \leq \frac{n^h}{h!}, \quad \forall n \geq h \geq 0 , \tag{A.67}$$

which also provides

$$\sum_{h=1}^{k} \binom{n}{h} \leq (n+1)^k . \tag{A.68}$$

Therefore, we can upper bound (A.62) using Lemma 3 as

$$\mathbf{E}_{\mathbf{g}}[t(\mathbf{g})^2] \leq \mathbf{E}_{\mathbf{g}}\left[ \max\left\{ \|\mathbf{g}_G\|_2^2 \ : \ G \in \mathcal{G}^{(k)}, G \subseteq S^c \right\} \right] \tag{A.69}$$

$$\leq \left( \sqrt{2 \log \left( (p - k - \left\lceil \tfrac{s}{k} \right\rceil + 2)^k \right)} + \sqrt{k} \right)^2 \tag{A.70}$$

and the statement follows. ∎

Now we are ready to prove the upper bound on the Gaussian width. First, note that

$$\omega(\mathcal{T}_{\mathcal{A}}(\boldsymbol{\theta}^*) \cap \mathbb{S}^{p-1})^2 \leq \mathbf{E}_{\mathbf{g}}[\mathrm{dist}(\mathbf{g}, \mathcal{N}_{\mathcal{A}}(\boldsymbol{\theta}^*))^2] \tag{A.71}$$

$$\overset{(a)}{\leq} \mathbf{E}_{\mathbf{g}}[\|\mathbf{g} - \mathbf{z}(\mathbf{g})\|_2^2] \tag{A.72}$$

$$= \mathbf{E}_{\mathbf{w}}[\|\mathbf{z}_S - \mathbf{g}_S\|_2^2] \tag{A.73}$$

$$\overset{(b)}{=} \mathbf{E}[\|\mathbf{z}_S\|_2^2] + \mathbf{E}[\|\mathbf{g}_S\|_2^2] \tag{A.74}$$

$$\overset{(c)}{=} \mathbf{E}[t(\mathbf{g})^2] \cdot \|\mathbf{v}_S\|_2^2 + |S| \tag{A.75}$$

$$\overset{(d)}{\leq} \left( \sqrt{2k \log \left( (p - k - \left\lceil \tfrac{s}{k} \right\rceil + 2) \right)} + \sqrt{k} \right)^2 \cdot \left\lceil \tfrac{s}{k} \right\rceil + s , \tag{A.76}$$

where $(a)$ follows from the definition of distance to a set, $(b)$ follows from the independence of $\mathbf{g}_S$ and $\mathbf{g}_{S^c}$, $(c)$ follows from the fact that the expected length of an $|S|$ length random i.i.d. Gaussian vector is $\sqrt{|S|}$, and $(d)$ follows since $|S| = \frac{ks}{k}$, and that $\|\mathbf{v}_S\|_2 \leq \sqrt{\left\lceil \tfrac{s}{k} \right\rceil}\|\mathbf{v}_S\|_k^{sp^*} = \sqrt{\left\lceil \tfrac{s}{k} \right\rceil}$. Thus inequality (28) follows. ∎

Next, we prove inequality (27). Let us denote $\mathbf{t} = \mathbf{X}^T \mathbf{w}$, and note that $\mathbf{t} \sim \mathcal{N}(0, I_p)$

$$\|\mathbf{X}^T(\mathbf{y} - \mathbf{X}\boldsymbol{\theta}^*)\|_k^{sp^*} = \|\mathbf{X}^T\mathbf{w}\|_k^{sp^*} = \|\mathbf{t}\|_k^{sp^*} = \max\{\|\mathbf{t}_G\|_2 \ : \ G \in \mathcal{G}^{(k)}\} . \tag{A.77}$$

Therefore, we can use Lemma 3 in order to bound the expectation $\mathbf{E}[\|\mathbf{t}\|_k^{sp^*}]$ as

$$\mathbf{E}[\|\mathbf{t}\|_k^{sp^*}] = \mathbf{E}[\max\{\|\mathbf{t}_G\|_2 \ : \ G \in \mathcal{G}^{(k)}\}] \tag{A.78}$$

$$= \mathbf{E}[\max\{\|\mathbf{t}_G\|_2 \ : \ G \in \mathcal{G}^{(k)}, \ |G| = k\} \tag{A.79}$$

$$\leq \left( \sqrt{2\log\binom{p}{k}} + \sqrt{k} \right)^2 \tag{A.80}$$

$$\leq \left( \sqrt{2k\log\left(\frac{ep}{k}\right)} + \sqrt{k} \right)^2 \tag{A.81}$$

where we have used the inequality

$$\binom{p}{k} \leq \left(\frac{ep}{k}\right)^k \tag{A.82}$$

Therefore, inequality (27) follows, and by our choice of $\lambda_p$, with high probability, $\boldsymbol{\theta}^*$ lies in the feasible set.

∎