[Reviews · NeurIPS 2014]

Submitted by Assigned_Reviewer_3

The paper provides a generalization of the Dantzig selector. My main concern about this paper is the motivation to study this problem. If there is a good motivation to study such problem, then I believe this paper can be accepted.

Otherwise, their main theorem is essentially a simple application of Gordon's escape through the mesh theorem which is well-known in the sparse recovery literature. Their algorithm is also a straightforward application of ADMM and hence is simple to derive. But again, if the authors provide a very good reason/application for studying such problems, the lack of theoretical novelty can be accepted.
Summary: My main concern about this paper is the motivation to study this problem. If there is a good motivation to study such problem, then I believe this paper can be accepted. The theoretical novelty of the paper is very limited.

Submitted by Assigned_Reviewer_25

This paper proposes ADMM-type algorithms for L0-regularized least squares problems, implementing the standard L0-based compressed sensing reconstruction criterion. The proposed algorithms are analyzed and theoretical guaranteed are provided, which depend on RIP properties of the system matrix. This is a very good quality paper, well written, and with a solid experimental evaluation.
Summary: A very interesting paper, which should be of interest to the NIPS community.

Submitted by Assigned_Reviewer_34

The paper introduces a generalization of the Dantzig selector, where the L1/Linfinity norm pair are replaced with an arbitrary norm R and its dual, along with supporting statistical theory and an inexact ADMM. The authors also provide detailed derivations of both proximal operators and statistical theory for the special case where R is the k-support norm.

This is very nicely written paper on an interesting subject. My only critical comment is that its motivation, beyond generalization for its own sake, is unclear. Why do we need a generalized Dantzig selector when there are already many papers on penalized loss methods? This is an important question, but I admit not completely fair to the authors---they're not alone in avoiding this question. Some ways to address this question include comparisons on real data and simulations, and/or commentary on the computational complexity and practical differences between applying this method and some other penalized regression method.

If we ignore the question of motivation and accept that a generalization of the Dantzig selector is worthwhile, then I believe that this work presents a significant advance in a very clearly written paper. Aside from the Dantzig selector, the auxiliary results on implementing the ADMM algorithm, including derivation of the proximal operator for the k-support norm, are worthwhile on their own.

Minor comment:

Inequality (27) of Theorem 4 is is contained in Lemma 3.3 of Gordon et al. (2007) which is an even stronger result. I believe that a bound similar to inequality (28) could also be derived using the results of Gordon et al. (2007).

Gordon, Y., Litvak, A. E., Mendelson, S., & Pajor, A. (2007). Gaussian averages of interpolated bodies and applications to approximate reconstruction. Journal of Approximation Theory, 149(1), 59–73. doi:10.1016/j.jat.2007.04.007
Summary: If we ignore the question of motivation and accept that a generalization of the Dantzig selector is worthwhile, then I believe that this work presents a significant advance in a very clearly written paper. Aside from the Dantzig selector, the auxiliary results on implementing the ADMM algorithm, including derivation of the proximal operator for the k-support norm, are worthwhile on their own.
Author Feedback
Author rebuttal: Reviewer#25:
We thank the reviewer for the encouraging comments.

Reviewer#3:
We thank the reviewer for the comments and would like to address concerns raised by the reviewer regarding the motivation of the work.

We have two main motivations behind our work. One is to provide a generalization of Dantzig Selector [3], originally designed for the L1 norm, to all norms. Note that the regularization in our work is based on the dual norm, which is more in spirit of the original Dantzig Selector, and is different from other recent advances in constrained estimation [4]. The second motivation is the analysis of the recently proposed k-support norm [1]. The k-support norm enjoys similar statistical properties as the elastic-net [19] which, unlike Lasso, is effective in selecting correlated relevant covariates. Although the k-support norm was introduced in [1] (NIPS’12) and studied in other papers [9], the recovery guarantee was left as an open question [1]. Further, there is no rigorous recovery guarantee for the elastic-net either. We establish recovery guarantees for the k-support norm as a special case of our Generalized Dantzig Selector (GDS) framework. On optimization side, we show that the GDS can be efficiently executed as long as the proximal operator for the norm (or its conjugate function) can be efficiently computed. In particular, for the k-support norm, we find a new algorithm to compute the proximal operator via its conjugate function. Note that the computation and analysis of the proximal operator of the conjugate of the k-support norm is non-trivial, is quite different and faster than previous work on k-support norm [1][9].

Reviewer#34:
We thank the reviewer for the encouraging comments.

As we outline above (response to Reviewer#3), the motivation is both to study the Generalized Dantzig Selector for any norm, including both computational and statistical aspects, as well as establish concrete results for the recently proposed k-support norm [1], where we show precise recovery guarantees and introduce an efficient algorithm for computing the proximal operator based on the conjugate function of the norm.

We thank the reviewer for the pointer to Gordon et al. (2007). It indeed looks highly relevant, especially for (27). We will also investigate if our analysis for (28) can be simplified based on these results. We revisited our proof in light of the tighter bounds in Gordon et al., and a simple modification in our proof is all we needed to match the order of the bounds. Specifically, we indeed obtain a term of the form k*log(p/k), which matches with the inequality order in Lemma 3.3 of Gordon et al. We obtain a similar improvement in tightening the bound for (28). In the final draft of the paper, we will cite Gordon et al.’s paper, and update our results with the tighter bounds.